# Wind tunnel experiments to quantify the effect of aeolian snow transport on the surface snow microstructure

Benjamin Walter[1], Hagen Weigel[1], Sonja Wahl[2], and Henning Löwe[1]

[1]WSL Institute for Snow and Avalanche Research SLF, Davos, Switzerland
[2]School of Architecture, Civil and Environmental Engineering, École Polytechnique Fédérale de Lausanne (EPFL), Lausanne, Switzerland

**Correspondence:** Benjamin Walter (benjamin.walter@slf.ch)

**Abstract.** The evolution of the surface snow microstructure under the influence of wind during precipitation events is hardly understood but crucial for polar and alpine snowpacks. Available statistical models are solely parameterized from field data where conditions are difficult to control. Controlled experiments which exemplify the physical processes underlying the evolution of density or specific surface area ($SSA$) of surface snow under wind are virtually non-existing. As a remedy, we conducted experiments in a cold laboratory using a ring-shaped wind tunnel with an infinite fetch to systematically investigate wind-induced microstructure modifications under controlled atmospheric, flow and snow conditions, and to identify the relevant processes. Airborne snow particles are characterized by high-speed imaging, while deposited snow is characterized by density and $SSA$ measurements. We used a single snow type (dendritic fresh snow) for simulating different precipitation intensities, vary wind speeds at a height of 0.4 m from 3 ms$^{-1}$ to 7 ms$^{-1}$ (for fixed temperature) and vary temperatures from -24°C to -2°C (for fixed wind speed). The measured airborne impact trajectories confirm the consistency of our coefficient of restitution with large scale saltation, rendering the setup suitable to realistically study interactions between airborne and deposited snow. Increasing wind speeds resulted in intensified densification and stronger SSA decreases. The most drastic snow density and SSA changes of deposited snow are observed close to the melting point. Our measured densification rates as a function of wind speed show clear deviations from existing statistical models but can be re-parameterized through our data. This study, as a first of its kind, exemplifies a rich non-linear interplay between airborne and deposited snow particles which is discussed in view of a multitude of involved processes, i.e. airborne metamorphism, cohesion, particle separation and fragmentation.

## 1 Introduction

The topmost centimeters of a snowpack (herein referred to as "surface snow") forms the interface to the atmosphere and is frequently affected by wind (e.g., Seligman, 1936; Mott et al., 2018). During a snowstorm, the typically dendritic precipitation particles with diameters of 1-5 mm (Woods et al., 2008; Garrett and Yuter, 2014) are subject to aerodynamic drag forces. Depending on wind speed and particle size and shape, the snow particles either roll on the ground (rolling or creeping), follow near ground ballistic trajectories occasionally impacting on the ground (saltation), or are transported without contacting the ground (suspension) (e.g., Walter et al., 2014). Snow particle fragmentation upon surface collisions during saltation (Sato et al., 2008; Comola et al., 2017), rounding due to abrasion while rolling, and sublimation in sub-saturated air (Dai et al., 2014)

are key factors determining the size, shape and packing density (Golubev and Sokratov, 2004; Cho et al., 2006) of the ulti-
mately deposited snow. Blowing snow particles are typically smaller than precipitation crystals (Schmidt, 1982; Nishimura
and Nemoto, 2005; Nishimura et al., 2014), with diameters ranging from 50–500 $\mu$m and with a higher sphericity and lower
dendricity (Bartlett et al., 2008). The deposited snow particles define the microstructure and thereby the properties of surface
snow like albedo, density, or cohesion. These properties are relevant for the mechanical stability of wind slabs for avalanche

formation (Schweizer et al., 2003), the exchange of chemical substances with the atmosphere (e.g., Pomeroy and Jones, 1996),
alpine and polar mass balances (Rignot and Thomas, 2002), or radiative transfer (Flanner and Zender, 2006). Thus, it is critical
for models of different spatial resolution from process based snowpack models to global scale climate models to accurately
simulate how the physical properties of the snow surface change under the influence of wind.

      The physical properties of snow are mainly dominated by two snow microstructural parameters: i) the snow density $\rho_s = \rho_i$

$\phi_i$, where $\rho_i = 917 \, \text{kg m}^{-3}$ is the density of ice and $0 < \phi_i < 1$ is the ice volume fraction of snow, and ii) the specific surface
area ($SSA$) defined as the surface area per ice mass (or volume) (e.g., Proksch et al., 2015; Warren, 2019). Surface snow
densities range from 70-100 $\text{kg m}^{-3}$ for Alpine fresh snow deposited under weak wind conditions, up to typically 250-400
$\text{kg m}^{-3}$ for strongly wind affected surface snow in Arctic and Antarctic regions (Brun et al., 1997; Fausto et al., 2018; Domine
et al., 2021). The $SSA$ of surface snow ranges from 70-150 $\text{mm}^{-1}$ for fresh snow (Yamaguchi et al., 2019) to 20-40 $\text{mm}^{-1}$ for

strongly wind affected snow in Antarctica (Gallet et al. 2010). For stationary surface snow under isothermal temperature con-
ditions, the density increases and the $SSA$ decreases over time (Schleef et al., 2014a). Wind generally results in an increase of
the surface snow density (Seligman, 1936, Sokratov and Sato 2001, Liston et al. 2007, Vionnet et al., 2012) with densification
rates that are one to two orders of magnitude higher than for isothermal metamorphism (Liston et al. 2007). Available statistical
models (Lehning et al., 2002, Liston et al. 2007, Vionnet et al., 2012) for the increase of surface snow density due to wind

are exclusively based on field measurements that are difficult to control and limited by the problem of accurately measuring
snow transport and microstructure simultaneously. Recently, Domine et al. (2019) found that the snow models SNOWPACK
(Lehning et al., 2002) and Crocus (Vionnet et al., 2012) greatly underestimate the increase in snow density for Arctic surface
layers. However, Royer et al. (2021), Wever et al. (2022) and Amory et al. (2021) illustrate how the parameterizations of the
increase of surface snow density due to wind can be adjusted to better represent the properties of surface snow in the Arctic

and in Antarctica. Cabanes et al. (2003) found that wind increases the rate of $SSA$ decrease, but insufficient data prevented a
parametrization of this effect. To the best of our knowledge, no other study yet investigated the effect of wind on the $SSA$.

      Different physical processes occurring during aeolian snow transport like particle fragmentation, sublimation and water va-
por re-deposition, may be responsible for the increase in deposited snow density and $SSA$ changes. Snow densification by
wind is believed to be mainly the consequence of particle fragmentation resulting in higher packing densities of the ultimately

deposited blowing snow particles (Sato et al., 2008; Comola et al. 2017). Fragmentation of snowflakes may already occur at
low wind velocities (Sato et al., 2008). For wind velocities < 2 m s$^{-1}$ measured at a height of 1 m, highly dendritic precipi-
tation snowflakes were found to not break upon collision with the surface, whereas they are completely decomposed for wind
velocities > 5 m s$^{-1}$. The number of fragments increased while the fragment size decreased with impact velocity (Sato et al.,
2008). A discrete element model (DEM) of the fragmentation process was presented by Comola et al. (2017), linking the size

distribution of blowing snow particles to that of falling snow crystals. Pure sublimation of windblown snow particles results in a reduction of snow particle size, and partial disappearance of entire grains, potentially resulting in a significant mass loss (Groot-Zwaaftink et al., 2013, Palm et al. 2017). A decreasing $SSA$ with increasing relative humidity was found by Yamaguchi et al. (2019) for precipitation particles, likely being the result of vapor deposition and riming. To discern the different processes responsible for snow density and SSA changes due to wind, it is necessary to complement field measurements by controlled laboratory experiments.

It is the aim of our study to propose an experimental setup to systematically investigate how wind affects the evolution of the surface snow density and $SSA$ during precipitation events as functions of the wind speed, air temperature and transport duration. We therefore deployed a ring-shaped wind tunnel (RWT) in a walk-in cold room to combine flow measurements (velocity, temperature, humidity) and particle characteristics (coefficient of restitution, impact angle and velocity) with established snow microstructure measurements (X-ray tomography, IceCube, density cutter) under controlled cold laboratory conditions. The relevant physical processes responsible for the density and $SSA$ changes, i.e., particle fragmentation, sublimation and vapor re-deposition are characterized, discussed and linked to the surface snow microstructure modifications.

The manuscript is organized as follows. The methods, instrumentation and models used and tested in this study are introduced in Section 2. The density and $SSA$ measurements, high-speed camera particle imaging, and the microstructural modifications due to wind are introduced, discussed, and compared to the models in Section 3. A summary of the results, the main conclusions and an outlook can be found in Section 4.

## 2  Methods

### 2.1  Ring wind tunnel

Experiments using a closed-circuit RWT were performed under stable temperature conditions in one of the WSL/SLF cold labs in Davos (Switzerland). The RWT (Fig. 1a) has an obround shape with two straight sections providing space for measurements without centrifugal effects as they occur in the curved sections. The RWT system is 2.2 m long and 1.2 m wide, and its tunnel is 20 cm wide and 50 cm high (Sommer et al., 2017). A model-aircraft propeller driven by an electric motor is used to create the airflow. The propeller is controlled by the wind velocity $V_{0.4m}$ measured at a height of 0.4 m to maintain a target wind velocity. In the empty RWT without snow, the maximum possible wind speed is approximately $V_{0.4m} \approx 8.5$ ms$^{-1}$. The infinite fetch of the closed-circuit RWT enables continuous snow transport with particles rolling or saltating on the ground, or being in suspension for longer time periods. This is not possible in open-circuit wind tunnels like the large-scale wind tunnel at WSL/SLF (e.g., Walter et al., 2012a-c), where particles are blown out of the tunnel after a few seconds of transport.

### 2.2  Drift experiments

We performed a total of 14 RWT experiments for testing the effects of wind speed (experiment 1-7), transport duration (experiments 8-9) and temperature (experiments 10-12) on the surface snow microstructure (Table 1). Particle impact characteristics

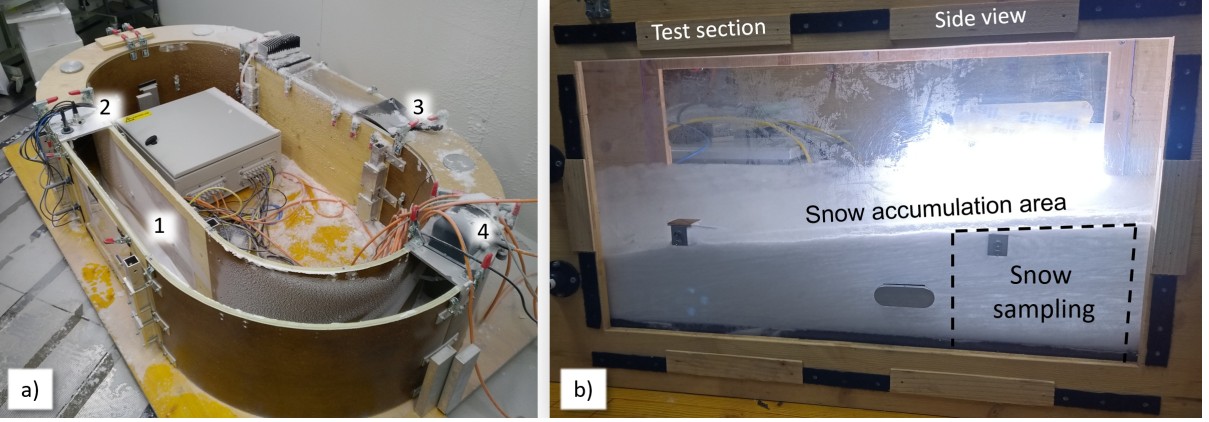

**Figure 1.** a) RWT installed in the cold laboratory at $T_a$ = -20°C: 1) Straight test section, 2) Measurement section with snow and air temperature, relative humidity, and wind speed sensors, 3) Inlet for adding fresh snow and 4) Wind turbine. The RWT ceiling is removed at the test section and at a part of the lower curved section in this image. The flow direction is counter clockwise. b) Side view of the test section with the snow accumulation area where the snow microstructure measurements were performed.

were measured with a high-speed camera during experiment 13, while experiment 14 served as a sensitivity study to test the effect of sublimation and vapor re-deposition on airborne particles. For the drift experiments, dendritic fresh snow was produced in the WSL/SLF snowmaker (Schleef et al., 2014b) at an air temperature of -20°C and a water bath temperature of +30°C, resulting in initial snow densities of 45-80 kgm$^{-3}$ and $SSA$ of 45-70 mm$^{-1}$ for the experiments. For each of the twelve main experiments, a constant wind velocity was initially set ranging in between $V_{0.4m}$ = 4 - 7.1 ms$^{-1}$ (Table 1). A total volume of (60 x 40 x 40 cm$^3$) of fresh snow was then slowly added to the wind tunnel, temporally equally distributed over the entire experiment duration $\tau_{exp}$ (Table 1), to mimic snow precipitation until the end of the experiment. The new snow was carefully poured into the wind tunnel at an inlet (Fig. 1a) with a shovel, minimizing the destruction of the initial microstructure. Only the microstructure of the snow crystals getting into contact with the shovel may be slightly affected by the pouring procedure, while their quantity is negligible compared to the entire volume of snow on the shovel. No sieving was applied to avoid preceding fragmentation of the highly dendritic snow crystals. Once a portion of new snow was poured into the RWT, it got immediately suspended and redistributed in the flow. The wind turbine propeller is located beneath the lid in the curved section of the ring wind tunnel (Fig. 1a), covering approximately the top quarter of the wind tunnel cross-section. The propeller blades are 90 mm long. As the particle mass flux exponentially decreases with height as shown by Yu et al. (2023) for our RWT, only a negligible part of the snow particles will get into contact with the propeller. A schematic drawing of the RWT and additional figures can be found in Yu et al. (2023).

The new snow in the Snowmaker box that served as the initial condition for the experiments, is regarded as the microstructure representing precipitation deposition unaffected by wind ($V_{0.4m}$ = 0 m s$^{-1}$). The settings and atmospheric conditions for the twelve experimental runs are summarized in Table 1. For the experiments 1-7, the wind velocity ranged from $V_{0.4m}$ = 4 − 7.1 m s$^{-1}$ while the transport duration (0.5 h) and the air temperature ($\approx$ -22 $\pm$1°C) were constant. For the experiments 8-12,

**Table 1.** Overview of the experimental settings and atmospheric conditions for the main experiments (1-12) and the complementary experiments (13-14). The average value for $RH$ measured with respect to ice is calculated from the second period of each experiment where a situation close to equilibrium for $RH$ is reached (Fig. 2).

| Experiment | Mean wind speed $V_{0.4m}$ $[ms^{-1}]$ | Experiment duration $\tau_{exp}$ $[h]$ | Average air temperature $T_a$ [°C] | Average relative humidity $RH$ [%] | $\mu CT$ measurements yes / no |
|---|---|---|---|---|---|
| 1 | 5.0 | 0.5 | -24.0 | 92.0 | no |
| 2 | 6.9 | 0.5 | -24.6 | 99.5 | no |
| 3 | 6.0 | 0.5 | -23.8 | 99.5 | no |
| 4 | 7.1 | 0.5 | -21.3 | 99.5 | no |
| 5 | 4.0 | 0.5 | -20.6 | 98.6 | yes |
| 6 | 6.6 | 0.5 | -20.6 | 98.7 | yes |
| 7 | 5.0 | 0.5 | -23.1 | 98.5 | yes |
| 8 | 6.0 | 1.0 | -21.7 | 98.1 | yes |
| 9 | 6.0 | 2.5 | -21.0 | 100.7 | yes |
| 10 | 6.0 | 0.5 | -11.5 | 100.5 | yes |
| 11 | 6.0 | 0.5 | -5.6 | 99.9 | yes |
| 12 | 6.0 | 0.5 | -2.4 | 99.4 | yes |
| 13 | 3.0 - 7.0 | 5.8 | -20.6 | 83.5 | no |
| 14 | 7.9 | 2.5 | -18.0 | 98.5 | yes |

the wind velocity was set to $V_{0.4m}$ = 6 m s$^{-1}$ while the experiment duration was varied ($\tau_{exp}$ = 1 h and 2.5 h, experiments 8 and 9) or the air temperature was changed to $T_a$ = -12°C, -6°C and -2°C (experiments 10 – 12). The major snow accumulations analyzed with the instruments discussed in the following Section developed in the straight test section of the RWT (Fig. 1b).

Two complementary experiments were conducted to i) measure and analyze particle impact characteristics and ii) to separately investigate the effect of long snow transport durations. Both experiments were conducted at an air temperature of $T_a \approx$ -20°C while a low mass of snow (600g) was poured into the RWT at the beginning of the experiment. For experiment i) The wind speed was varied from $V_{0.4m}$ = 3 – 7 m s$^{-1}$ and near surface particle impacts were recorded with a high-speed camera (Experiment 13 in Table 1). A total of 75 particle impacts were analyzed for the different wind velocities. For experiment ii) the wind speed was set to $V_{0.4m}$ = 8 m s$^{-1}$ to keep the particles in suspension as long as possible, and the transported particles were sampled out of the air every 15 min (Experiment 14 in Table 1).

## 2.3 Instrumentation and measurements

The straight test section (Fig. 1a and b) is equipped with various sensors for continuously measuring the air temperature $T_a$ (HC2-S, Rotronic, uncertainty: ±0.3°C), the relative humidity $RH$ (HC2-S, Rotronic, uncertainty: ±1.5%) and the wind speed

$V_{0.4m}$ (MiniAir, Schildknecht, uncertainty: $\pm 0.2$ ms$^{-1}$). The air temperature and relative humidity are measured at a height of 0.15 m, while the wind speed is measured at a height of 0.4 m above the wind tunnel floor. Because the snow height increased during the experiments, the height of the sensors above the snow surface decreased, however, none of the sensors was covered in snow at anytime. The atmospheric data is sampled at a frequency of 5 Hz using a custom-made software based on LabVIEW (National Instruments). A high-speed camera (Phantom, Vision Research) is used to record particle impacts close to the snow

surface and to calculate particle impact characteristics like the coefficient of restitution and impact and ejection angles.

   Snow microstructure measurements were performed before and after the drift experiments to characterize the fresh and the wind affected snow microstructures. An IceCube (A2 Photonics Sensors) instrument was used to measure the specific surface area of the snow in the measurement section (Fig. 1b) with a stated measurement uncertainty of $\pm 10\%$ (Gallet et al., 2009). Recently reported systematic IceCube measurement errors detected for SSA $< 20$ mm$^{-1}$ (Martin and Schneebeli et al., 2023)

are unproblematic for our dendritic fresh snow experiments with SSA $> 20$ mm$^{-1}$. A standard density cutter (5.5 x 6 x 3 cm$^3$) was used to measure the snow density. The cutter and $\mu$CT density measurements were found (Proksch et al., 2016) to agree within 5-9%, thus resulting in a similar overall uncertainty for the density and for the $SSA$ measurements. Up to 30 IceCube and density cutter measurements were performed for each experiment depending on the dimensions of the snow accumulation to characterize the spatial variability of the snow density and $SSA$.

Micro-computed tomography ($\mu$CT) measurements of the initial and final snow microstructures were performed for the experiments 5-12 to access information on particle size distributions and to obtain complementary measurements of the $SSA$ and the snow density. A cylindrical snow sample with a diameter of 36 mm and a height of 70 mm was vertically cut out of the initial new snow or the accumulated snow for the $\mu$CT measurements. Because of the limited sampling volume, only little information on the spatial density and $SSA$ variability is obtained from these measurements. The $\mu$CT measurements for

the main experiments (Experiments 1-12, Table 1) were performed with a Scanco CT-40 scanner with a voxel size of 18 $\mu$m (e.g., Pinzer and Schneebeli, 2009; Heggli et al., 2011). The rather low resolution is chosen because of the necessity of using sufficiently large sample holders for taking snow samples from the deposit. For smaller sample holders, an impact of sampling on the density is unavoidable for the densities of interest. Therefore it cannot be ruled out that absolute values of the $SSA$ are biased low due to resolution. A smaller sample holder was used for the additional Experiment 14 resulting in a voxel size of 8

$\mu$m. For the binary segmentation, the energy-based segmentation procedure presented by Hagenmuller et al. (2013) was used. A marching cubes approach (Hagenmuller et al., 2016) is used for calculating the $SSA$ from the segmented $\mu$CT images.

## 2.4  Densification models

To make contact to field-based results, three available model formulations for wind induced snow densification during snow precipitation events (SNOWPACK, Lehning et al. (2002), SnowTran-3D, Liston et al. (2007); CROCUS, Vionnet et al. (2012))

are tested in our study. In all three models, snow densification by wind can be active with and without concurrent snowfall and is initiated when wind speed exceeds certain thresholds to generate snow transport. Therefore, all models include terms describing ground-based densification of surface snow layers due to wind transport, and new snow density terms ($\rho_{ns}$) that also depend on wind speed for describing an initial compaction of precipitation. This separation is likely resulting from different

time scales and snow types involved in these different processes. During precipitation events, the typically highly dendritic new snow will quite quickly (depending also on the wind speed) cover the underlying (new) snow which then can't be entrained anymore by the wind, resulting in rather short effective transport durations. However, without precipitation, loose surface snow of likely different grain types (potentially more decomposed rounded particles) may be entrained and affect by wind for much longer time (or transport) durations. As we simulate precipitation events in our RWT, only the equations for the wind speed dependent new snow density $\rho_{ns}$ with concurrent snowfall are considered in our study. In Zwart (2007), a relationship between the wind velocity, air temperature, relative humidity and new snow density used in SNOWPACK was derived by relating local weather station data to snow density measurements in an alpine catchment:

$$\rho_{ns} = 10^{\beta_0 + \beta_1 T_a + \beta_2 \arcsin(\sqrt{RH}) + \beta_3 \log_{10}(V)} \tag{1}$$

In Eq. 1, $\beta_0$ = 3.28, $\beta_1$ = 0.03, $\beta_2$ = -0.75 and $\beta_3$ = 0.3 are constants, $T_a$ is the air temperature in [°C], $RH$ is the dimensionless relative humidity varying between 0 and 1, and $V$ the wind velocity at 2 m height in [m s$^{-1}$]. In contrast, Liston et al. (2007) describes the evolution of the surface snow density during periods of precipitation as a function of the temperature dependent new snow density $\rho_{ns}$ plus a wind-related density offset for wind speeds > 5 m s$^{-1}$:

$$\rho_{ns} = 50 + 1.7(T_{wb} - 258.16) + D_1 + D_2[1 - e^{-D_3(V-5)}] \tag{2}$$

In Eq. 2, $D_1$ = 25, $D_2$ = 250 and $D_3$ = 0.2 are constants, $T_{wb}$ the wet-bulb air temperature in [K] and $V$ the wind speed at 2 m height in [m s$^{-1}$]. Another empirical description for the new snow density affected by wind during precipitation events is implemented in the CROCUS snow model (Brun et al., 1997; Vionnet et al., 2012) which is also expressed as a function of the wind speed V and the air temperature $T_a$:

$$\rho_{ns} = a_\rho + b_\rho(T_a - T_{fus}) + c_\rho\sqrt{V} \tag{3}$$

In Eq. 3, $T_{fus}$ is the temperature of the melting point for water, $a_\rho$ = 109 kg m$^{-3}$, $b_\rho$ = 6 kg m$^{-3}$ K$^{-1}$ and $c_\rho$ = 26 kg m$^{-7/2}$ s$^{-1/2}$ are constants, and the minimal initial density is 50 kg m$^2$ (Pahaut, 1975).

## 3 Results and discussion

### 3.1 Atmospheric and flow conditions

The air temperature ($T_a$), relative humidity ($RH$), and wind speed ($V_{0.4m}$) are continuously measured and provide a basis for the surface snow microstructure analysis and discussion. The air temperatures typically increased by about 1-2°C during an experiment due to turbulent energy dissipation in the flow and heat generated by the turbine (Fig. 2). The RWT is not sealed, and air is exchanged between the RWT and the cold lab mainly at the inlet where the fresh snow was poured into the RWT (Fig. 1a). The relative humidity typically increased significantly by about 5-10% during the first 5-10 min of the experiments due to sublimation of snow particles in suspension and at the snow surface. Afterwards, an equilibrium situation is reached between sublimation and dry air exchange between the RWT and the cold lab at the inlet resulting in a rather stable or minor increase of

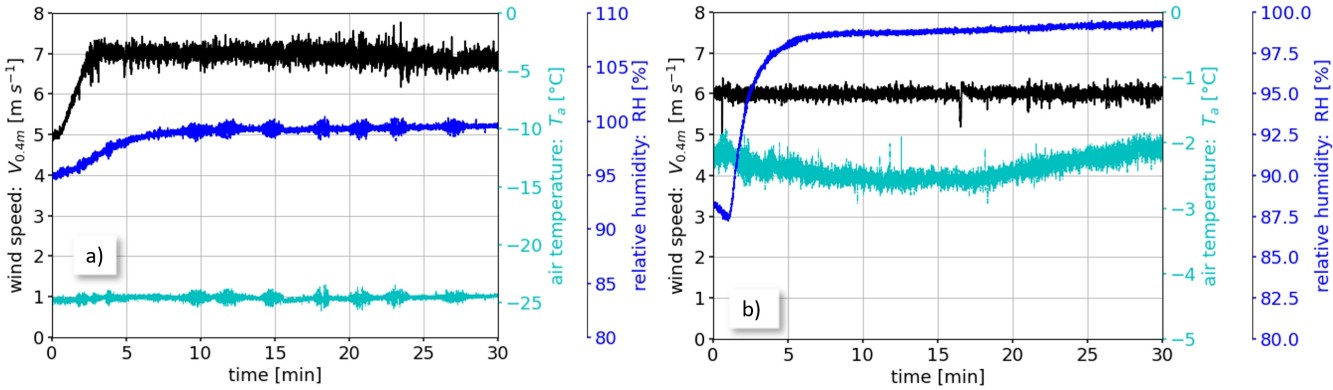

**Figure 2.** a) Evolution of the atmospheric conditions for experiment a) 2 and b) 12 including the wind speed ($V_{0.4m}$), air temperature ($T_a$) and relative humidity ($RH$).

$RH$ until the end of the experiment. In Fig. 2, $RH$ is shown with respect to ice and thus typically close to saturation during the second half of the experiments. The relative humidity of the cold room varies depending on how often people enter and leave the room during a day. In the morning, $RH$ is typically low at around 40% - 50%. Therefore, depending on the initial relative humidity before an experiment, a more or less strong increase in $RH$ is obtained due to sublimation of the suspended particles (Dai and Huang, 2014).

### 3.2 Particle transport characteristics

To compare the quality of our particle transport phase to previous studies of well-developed boundary layer flows, we address the consistency of particle transport and impact characteristics. A known limitation of the closed-circuit RWT is, that the deflection of the flow in the curved sections and the small tunnel cross-section prevents the development of a stable logarithmic boundary layer flow (Sommer et al., 2017) which is indeed critical for flow properties. In contrast, for this study, it is important that the relevant particle transport properties, i.e. the particle impact angles and velocities in the saltation layer are comparable to those of a fully developed natural boundary layer flow. These particle properties drive particle fragmentation, abrasion, and sublimation, and therefore the interaction with the microstructure of the ultimately deposited snow. We thus argue here that the boundary layer flow may not necessarily be perfectly homogeneous, stationary, and well-developed, as long as the particle impact characteristics are consistent with those of a well developed boundary layer flow as studied by Sugiura et al. (2000). Using the high-speed camera setup we measured particle impact angles, ejection angles and velocities for a wide range of different wind speeds (Experiment 13, Table 1).

An image sequence of a dendritic snow particle impacting on a snow surface recorded with a high-speed camera is shown in Fig. 3 as an exemplary case. The incoming particle merely rotates before the impact (Fig. 3a). The vertical ($V_{in,y}$) and horizontal ($V_{in,x}$) particle velocities define the absolute impact velocity $V_{in}$. Fig. 3b shows the situation at the time where the particle is in contact (impacting) with the stationary snow surface. The impact angle $\alpha_{in}$ and the ejection angle $\alpha_{out}$ are defined

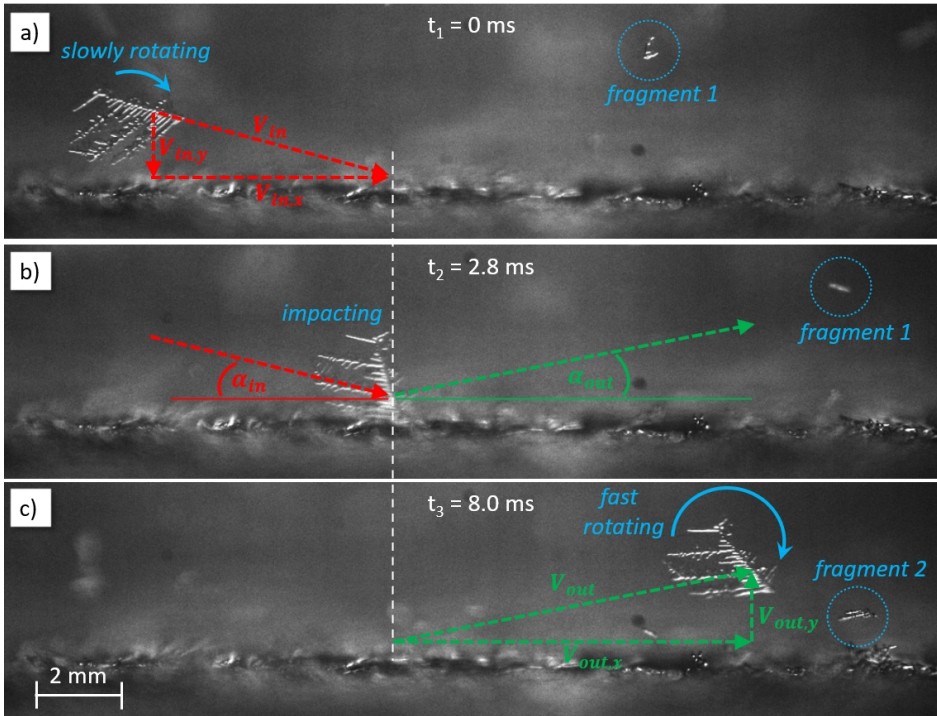

**Figure 3.** High-speed camera recording of a dendritic new snow particle impacting on a snow surface for an experiment with a wind speed of $V_{0.4m} = 6$ m s$^{-1}$. a) The particle slowly rotates before impacting at the surface with an impact velocity $V_{in}$. b) The particle impacts with the surface at an impact angle $\alpha_{in}$ and leaves the surface at an ejection angle $\alpha_{out}$. c) the particle is ejected at a different, lower ejection velocity $V_{out}$ and is fast rotating. Furthermore, two different fragments being transported by the wind are shown.

by the incoming and outgoing particle trajectories. Depending on the amount of energy that is dissipated or transformed on impact, the particle is ejected at a lower ejection velocity $V_{out}$ (Fig. 3c). In this case, no fragmentation of the impacting particle happened, however, the ejected particle is strongly rotating after the impact. Additionally, two different fragments that are transported by the wind are shown in the three images of Fig. 3.

The energy dissipation upon particle impact can be quantified through the apparent coefficients of restitution. The relevant collision quantities, the impact velocities ($V_{in} \approx 1\text{-}5$ m s$^{-1}$), the impact angles ($\alpha_{in} \approx 1\text{-}20°$) and the vertical ($C_v = V_{out,y}$ / $V_{in,y}$) and horizontal ($C_h = V_{out,x}$ / $V_{in,x}$) coefficients of restitution (Fig. 4) compare well with those measured by (Sugiura et al., 2000) in a large-scale boundary layer wind tunnel under well developed and stationary flow conditions. The values for the vertical coefficient of restitution are in the range of $C_v \approx 0.5\text{-}18$ (Fig. 4a) which are slightly higher than those found by Sugiura et al. (2000), who measured values for $C_v \approx 1\text{-}3$. The reason for this difference is our high-speed camera setup that allowed for high resolution images close to the ground (Fig. 3) and thus the detection of very low impact and ejection angles down to $\alpha_{in} = 1°$ (Fig. 4b), while the data of Sugiura et al. (2000) is limited to $\alpha_{in} > 5°$. Particles with low impact angles and high ejection angles result in high vertical coefficients of restitution $C_v$ (yellow dots in Fig. 4) and explain the difference to

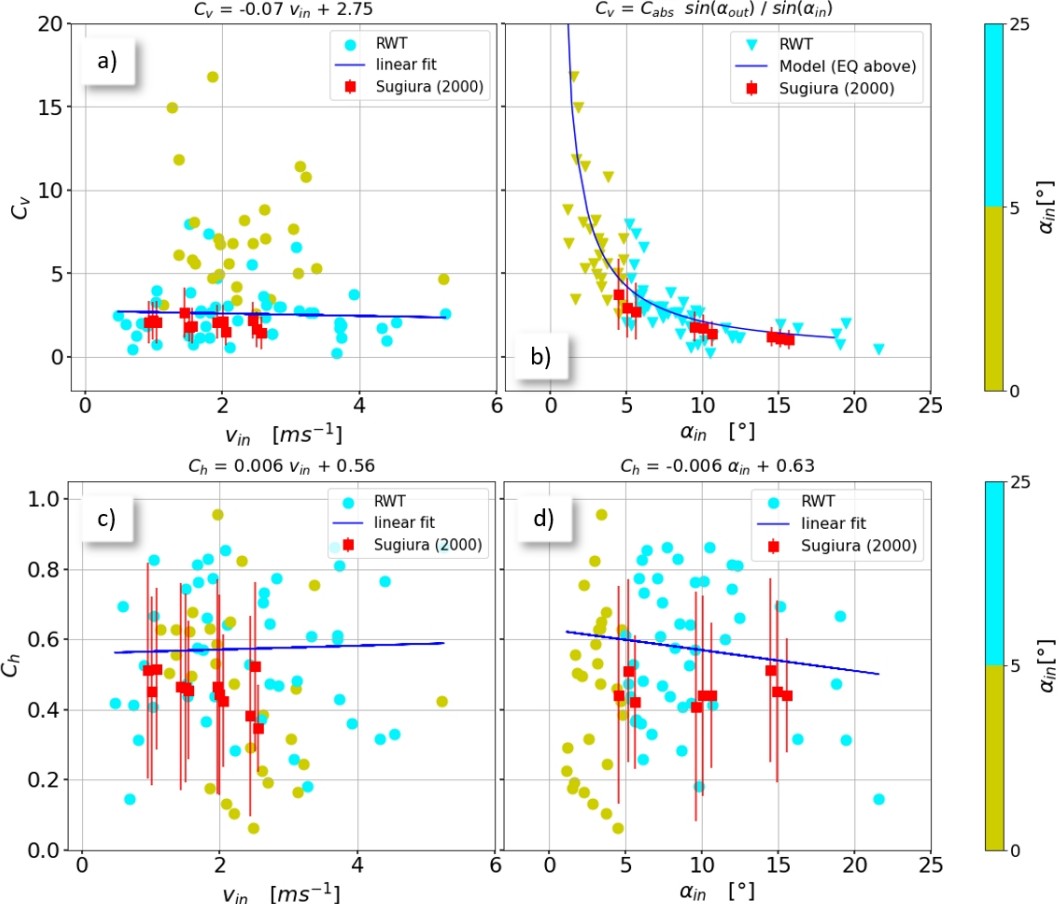

**Figure 4.** Vertical $C_v$ (a and b) and horizontal $C_h$ (c and d) coefficients of restitution as functions of the impact velocity $V_{in}$ (a and c) and the impact angle $\alpha_{in}$ (b and d). Note: Our RWT data is color coded to visualize the agreement with Sugiura et al. (2000) for $\alpha_{in} > 5°$.

Sugiura et al. (2000). For $C_v$ we find a clear dependence on the impact angle $\alpha_{in}$, similar to Sommerfeld et al. (2021), which can be cast into the form:

$$C_v(\alpha_{in}) = \overline{C_{abs}} \, \overline{\sin(\alpha_{out})} \, \sin^{-1}(\alpha_{in}) \tag{4}$$

Eq. 4 is plotted in Fig. 4b with $C_{abs} = |V_{out}|/|V_{in}|$ (Fig. 5b). For a better comparability of our horizontal and vertical coefficients of restitution with the results of Sugiura et al. (2000), the linear fits shown in Fig. 4a, c and d are limited to the collisions with $\alpha_{in} > 5°$. Our horizontal coefficients of restitution span the same range of $C_h \approx 0.1$ - $0.9$ (Fig. 4c-d) as those found by Sugiura et al. (2000). These collision quantities ($C_v$ and $C_h$) define the forces acting on the dendrites or branches of the fresh snow particles during an impact and thus the degree of fragmentation (Comola et al., 2018).

To complete this impact analysis, we show in Fig. 5a that the average impact velocity $V_{in}$ increases with increasing wind speed $V_{0.4m}$. Fig. 5b demonstrates that, despite considerable scatter, the absolute coefficient of restitution $C_{abs}$ slightly in-

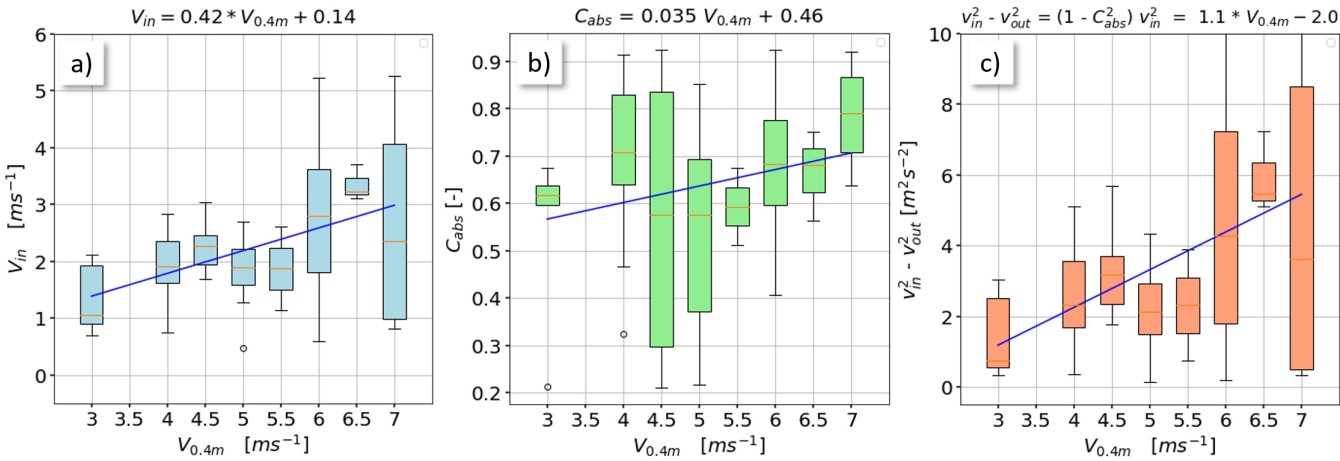

**Figure 5.** a) Particle impact velocity $V_{in}$, b) absolute coefficient of restitution $C_{abs}$ and c) normalized dissipated impact energy $V_{in}^2 - V_{out}^2$ as a function of the RWT wind speed $V_{0.4m}$ including linear fits.

creases with wind speed $V_{0.4m}$. This implies that the total energy available for the densification processes, the normalized dissipated impact energy $V_{in}^2 - V_{out}^2$ increases with wind speed (Fig. 5c). The p-values for the statistical significance are p = 0.011 (strong evidence, Fig. 5a), p = 0.079 (weak evidence or trend, Fig. 5b), and p = 0.0067 (strong evidence, Fig. 5c).

A visual inspection of the saltation layer shows a typical saltation layer height of $h_{salt} \approx 30 - 50$ mm with a homogeneous particle distribution across the RWT cross-section in the straight test section (not shown here). Comparable saltation layer heights were measured for well-developed natural boundary layer flows in the large-scale wind tunnel using shadowgraphy imaging (Gromke et al., 2014). The particle concentration in the RWT also exponentially decreases with height as shown by the measurements of Yu et al. (2023) while studying snow cornice formation in the same RWT. In the curved sections, a large portion of the snow particles are transported along the outer wall due to centrifugal forces. The smallest particles are transported in suspension without contacting the snow surface and are partially blown out of the wind tunnel at the inlet. Overall, the results of this Section emphasize that our boundary layer flow is sufficiently developed which substantiates the feasibility of quantifying the effect of wind on the surface snow microstructure using the RWT.

## 3.3 Snow density and $SSA$ measurements

An increase in snow density is measured for all experiments while the magnitude of the increase depends on the flow and atmospheric conditions (Fig. 6a-b). The horizontal and vertical density variability across the wind tunnel width was measured with the density cutter for 2-3 profiles in flow direction at the snow accumulation area (Fig. 1b), resulting in a total of 5-30 density measurements for each experiment depending on the amount and height of the snow accumulation. The initial new snow densities representing snow deposited without wind ($V_{0.4m} = 0$ m s$^{-1}$) measured inside the Snowmaker box were ranging in between 45-80 kg m$^{-3}$ (Fig. 6a-b) and define the initial ice-volume fraction $\phi_{i0}$ for the calculation of the densification rates in

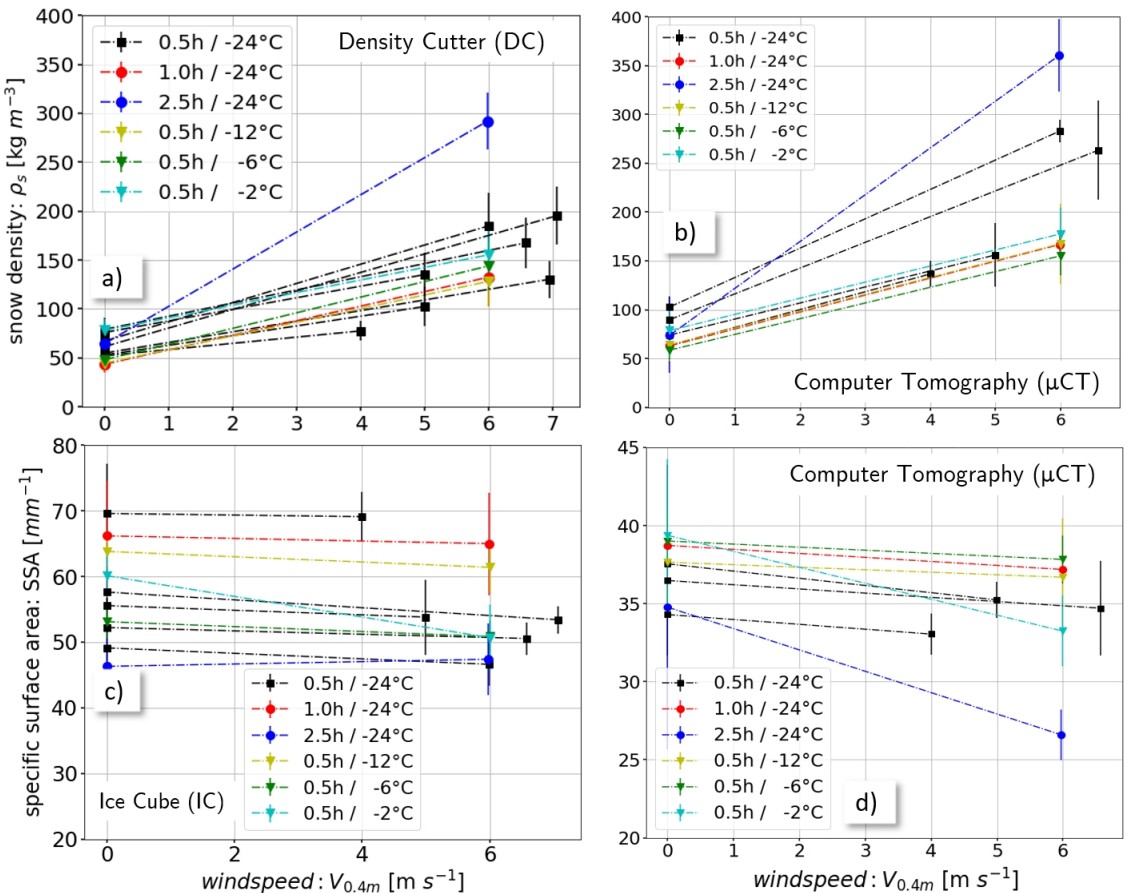

**Figure 6.** Summary of the snow densities $\rho_s$ measured with a) the density cutter and b) $\mu$CT, and the specific surface area $SSA$ measured with c) the IceCube instrument and d) $\mu$CT as a function of the windspeed $V_{0.4m}$. $V_{0.4m} = 0\ ms^{-1}$ corresponds to the initial new snow from the Snowmaker box. The error bars indicate one standard deviation of the variability (including spatial variability) of the measurements. A different y-scaling is used in d) relative to c) for a better visualization of the $SSA$ changes for the $\mu$CT measurements.

the following figures. These initial densities increased after wind exposure to values ranging from 75-350 kg m$^{-3}$, depending on the wind speed, air temperature and transport duration. An overall good agreement with strongly increasing densities for both, the cutter (Fig. 6a) and the $\mu$CT (Fig. 6b) measurements is found for the experiments 5-12 where $\mu$CT measurements are 
255 available (Table 1). However, differences of up to 20% between the individual experiments are attributed to general differences between the two methods as described in Section 2.3 and the spatial variability (Fig. 7) which is poorly represented by the single $\mu$CT sample of limited dimension.

The corresponding $SSA$ values measured with the IceCube and $\mu$CT are shown in Fig. 6c-d, both indicating a reduction in the $SSA$ values due to wind exposure. The initial new snow $SSA$ values are ranging from 45 mm$^{-1}$ < SSA < 70 mm$^{-1}$ for the 
260 IceCube measurements, whereas lower values of 35 mm$^{-1}$ < SSA < 40 mm$^{-1}$ were measured with the $\mu$CT. The lower $\mu$CT

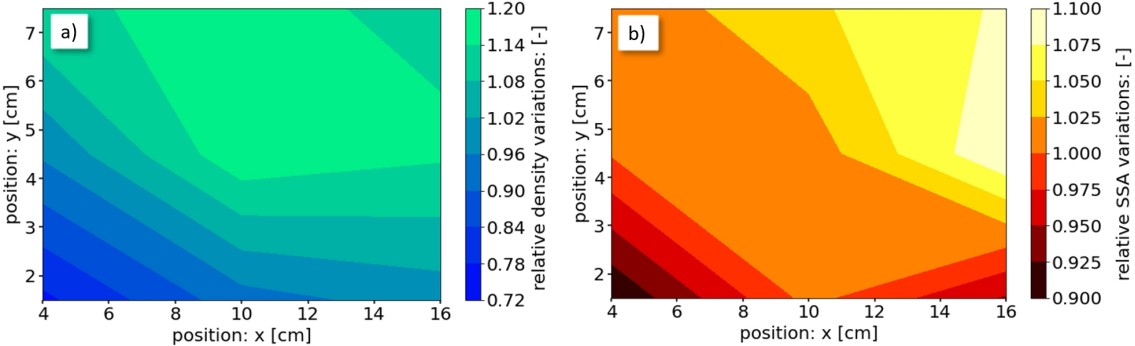

**Figure 7.** Average of the spatial variability of a) the snow density measured with the density cutter, and b) the $SSA$ measured with the IceCube instrument. Position $x = 4$ cm corresponds to the inside wall of the RWT, so the view is in flow direction.

values are partially the result of the $\mu$CT resolution with a voxel size of 18 $\mu m$ as discussed in Section 2.3. Differences between IceCube and $\mu$CT measurements may also be based on other reasons like the $\mu$CT $SSA$ computation algorithm or grain shape (Hagenmuller et al., 2016). However, the relative changes (e.g. $SSA$ rates in Fig 11b) are consistent between the IceCube and $\mu$CT measurements. Because the aim of this study is the investigation of relative changes in the snow density and $SSA$
induced by wind, the discrepancy in the absolute $SSA$ values is not necessarily a problem but must be considered carefully for the interpretation of the results. The error bars in Fig. 6 include, besides systematic and statistical measurement uncertainties, also the spatial variability (Fig. 7) of the snow density and $SSA$ in the accumulation area (Fig. 1b) for the density cutter and IceCube measurements. The error bars for the $\mu$CT measurements in Fig. 6b, d are derived from the standard deviation of the vertical density and SSA variability of the 3-6 cm high snow samples with 20-40 data points.
Fig. 7 shows the vertical and horizontal density and $SSA$ variations across the wind tunnel cross-section averaged for all experiments. The variability is first normalized by the mean for each experiment before an average of all 12 (10 for the SSA) experiments is calculated. The density varies about +/- 25% relative to the mean, with the lowest densities at the lower left of the vertical snow profiles which corresponds to the inside wall of the RWT at $x = 4$ cm. The wind velocities are lowest at this location where the first, thus least fragmented snow accumulations with rather low densities appear. While the snow
accumulations grow during the experiment, the densities tend to increase with height and towards the outer wall at $x = 16$ cm. Increasing densities with height can be explained by the gentle slope that forms during the experiment facing the wind (Fig. 1b). Sommer et al. (2018) found increasing densities with increasing slope angle of the snow cover because of steeper snow particle impact angles and thus most likely enhanced fragmentation and higher packing densities.

The spatial $SSA$ variability is overall smaller compared to the density with +/-10 % variation across the wind tunnel cross-
section (Fig. 7b). The smallest $SSA$ values are found at the lower left corner at $x = 4$ cm while the $SSA$ increases towards the upper right. One explanation could be that initially the smallest fragments sublimate due to the low relative humidity (Fig. 2) and that the largest particles from the added new snow accumulate first, resulting in initial snow accumulations with a

rather low $SSA$. However, the measured spatial $SSA$ variability is of the same order of magnitude as the $SSA$ measurement uncertainty (Section 2.3).

## 3.4 Snow densification rate

The snow densification rate describes the temporal increase in density relative to the initial snow density and is defined as $\phi_{rate}$ = $(\phi_i - \phi_{i0}) / (\phi_{i0} \Delta t)$, where $(\phi_{i0})$ is the initial and $(\phi_i)$ the final ice volume fraction after a time duration $\Delta t$ where snow transport occurs. To understand the origin of snow densification we consider the kinetic energy loss by impacting particles which can only be transformed into rotational kinetic energy (Fig. 3), friction (heat), bed compression, or abrasion and fragmentation. Rotation and friction (heat) cannot be responsible for densification. The pressure that saltating particles exert on the snow surface resulting in bed compression can be estimated from the particles momentum difference before and after the impact. An upper limit for this pressure estimate includes the assumptions of ice spheres of 0.5 mm diameter, purely vertical impacts, a maximum velocity difference of 3 m s$^{-1}$ before and after the impact, and 10 impacts per square centimeter and second. The maximum pressure estimate of 0.02 Pa is too low to result in a significant bed compression and thus snow densification when being compared to the pressure of 133 Pa that was applied to a snow surface during the compaction experiments of Schleef et al. (2014a). Therefore, fragmentation and abrasion resulting in smaller fragments (e.g. dendrites broken off from new snow crystals) and thus higher packing densities (Parteli et al., 2014) are assumed being the major processes behind wind induced surface snow densification. This conclusion is also supported by the findings of Sato et al. (2008), showing that snow flakes are completely decomposed when impacting a surface for wind speeds > 5 m s$^{-1}$ during precipitation.

### 3.4.1 Dependence on wind speed and involved mechanisms

Fig. 8 summarizes the densification rates for the experiments with different wind velocities (Table 1) which are derived from the averages of the density cutter and the $\mu$CT measurements weighted by their sampling volume. Furthermore, Fig. 8 includes model predictions and model fits to our RWT data using the theoretical descriptions from the SNOWPACK, SnowTran-3D and CROCUS models (Eq. 1-3). The densification rates for the experiments (black squares in Fig. 8) show an increase with increasing wind speed from $\phi_{rate} = 1.3 - 4.3$ h$^{-1}$. These rates are two to three orders of magnitude higher than those measured for isothermal metamorphism by Schleef et al. (2014a), which underlines the effect of wind.

Comola et al. (2017) showed with simulations that the average fragment size decreases and the number of fragments increases with increasing particle impact velocities $V_{in}$. As $V_{in}$ was found to increase with increasing wind speed $V_{0.4m}$ (Fig. 5a), we assume that also in our case the average fragment size decreases and the packing density increases with increasing wind velocity $V_{0.4m}$, thus being responsible for the observed densification rate increase shown in Fig. 8. Furthermore, stronger fragmentation at higher wind velocities requires higher impact energies being dissipated by plastic deformation of snow crystals. To test this assumption, and therefore also the simulated results of Comola et al. (2017), the dissipated impact energy is estimated by calculating the normalized kinetic energy difference $V_{in}^2 - V_{out}^2 = (1 - C_{abs}^2)V_{in}^2$ before and after the particle impacts. Fig. 5c shows that the dissipated impact energy increases with increasing wind velocity $V_{0.4m}$ (despite an increase of the absolute coefficient of restitution shown in Fig. 5b), likely resulting in smaller fragments, higher packing densities and thus

stronger densification. The quality and resolution of the high-speed camera recordings did, however, not allow for analyzing individual particle impacts on fragmentation.

In the curved sections, the particles are transported within a few centimeters distance from the vertical RWT outer wall. The visually identified modes of transport were a mixture of bouncing (saltation), rolling, and sliding along the wall, thus similar to saltation at the horizontal snow surfaces at the straight sections. The particle transport along the curved side walls is inevitable for a compact closed circuit wind tunnel in a cold laboratory of limited dimensions. The centrifugal forces acting on snow particles in the curved section were estimated being one to two orders of magnitude smaller compared to the forces acting on the snow particles during surface impact while saltating. The maximum centrifugal force was calculated as $F_c = m_p * v_p^2/r = 4.3$ $\mu$N for a large spherical snow particle of 0.5 mm diameter with a mass of $m_p = 0.06$mg, a maximum horizontal velocity of $v_p$ = 6 $ms^{-1}$ ($V_{0.4m} \approx 7\ ms^{-1}$) and the RWT radius of the curved section of r = 0.5 m. Horizontal snow particle velocities in snow saltation layers can be approximated as being about 1-2 m$s^{-1}$ lower than the mean horizontal wind speed (Nishimura et al., 2014). The maximum impact force can be calculated as $F_i = \Delta E_k/h = 360\ \mu$N, where $\Delta E_k$ is the kinetic energy difference before and after an impact of a similar snow particle of mass ($m_p = 0.06$mg) estimated from Fig. 5c as $\Delta E_k = 0.5*m_p*(V_{in}^2 - V_{out}^2) = 0.5 * m_p * 6\ m^2s^{-2}$ at the same wind speed of $V_{0.4m} = 7\ ms^{-1}$. An unknown parameter in this estimate is the height $h$ which defines the particle penetration distance into the snow surface. For small $h$ equal to the particle diameter, the particle impact force is about two orders of magnitude larger than the centrifugal force according to the values above. For increasing penetration distances $h$ (depending on the snow surface elastic or plastic deformation potential), the impact force decreases but is still one order of magnitude larger than the centrifugal force even for $h$ equal 8 times the particle diameter. We conclude that centrifugal forces in the curved section are negligible compared to surface impact forces for our RWT experiments.

Besides the centrifugal forces along the curved side walls, the first impact of snow particles into the vertical, curved side walls after the straight sections introduce additional unnatural mechanical stress on the snow particles, potentially affecting fragmentation. The above introduced estimate of the impact force $F_i$ onto the horizontal snow surface is based on impact characteristics determined from the particle tracking measurements, data that is not available for the first impacts at the curved sections. Therefore, we can only provide a discussion of potential differences that may in- or decrease the wall impact force relative to the snow surface impact force. The impact angles of the snow particles' first impact into the side walls were calculated (based on geometrical considerations) to be within a range of 5°-25°. These angles are comparable to the observed impact angles $\alpha_{in}$ on the horizontal snow surface in the straight test section (Fig. 4b and d). The maximum particle impact velocities into the side wall can again be estimated being 1-2 m$s^{-1}$ lower than the mean horizontal wind speed, thus about $v_p$ = 5-6 $ms^{-1}$ ($V_{0.4m} \approx 7\ ms^{-1}$). These maximum impact velocities are comparable to the maximum impact velocities $V_{in}$ on the horizontal snow surface (Fig. 5a). Geometric vector analysis revealed similar wall normal velocity components for the snow and the curved wall impacts. While the impact angles and velocities are similar, the hard wooden surface of the curved side walls likely increases the impact force relative to the snow surface. Contrarily, the smooth surface of the side walls is assumed to reduce the ejection angle and increase the ejection velocity compared to a snow surface impact, resulting in a decrease of the normalized dissipated impact energy (Fig. 5c) and impact force. The impact angle and the impact force may further be reduced by the particles' ability to follow the flow. Smaller particles (< 100 $\mu m$) have a good flow following behavior (Huang et al.,

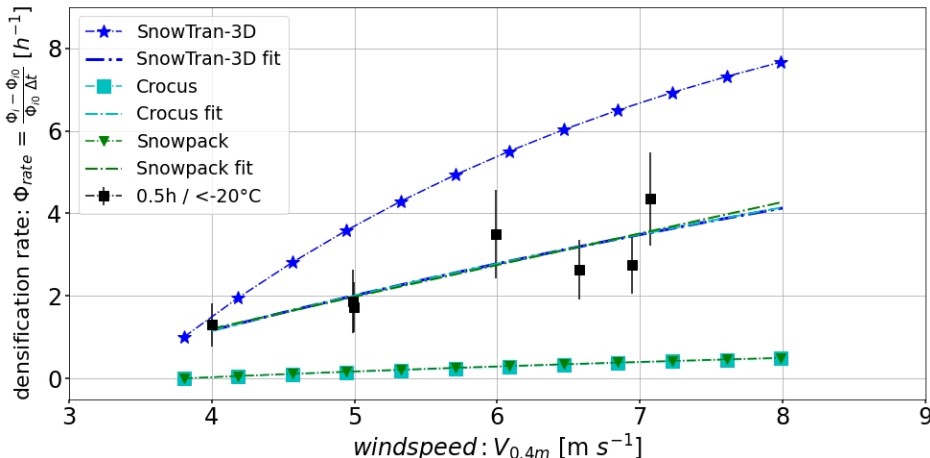

**Figure 8.** Densification rate as a function of wind speed for the Experiments 1-7 with constant air temperature ($T_a \approx$ -22°C) and experiment duration ($\tau_{exp}$ = 0.5 h) including the results of the SnowTran-3D (Liston et al., 2007), the SNOWPACK (Lehning et al., 2002) and the CROCUS (Vionnet et al. 2012) model predictions and model fits to our experiments.

2015) resulting in a reduction of the impact angles and thus forces. Vice versa, larger particles (> 300 $\mu m$) have a poor flow following behavior resulting in a minor reduction of the impact angle and force. An estimate of the particle size distribution for our experiments (Section 3.6.1, Fig. 12a) reveals that the majority of our snow particles are of a size smaller than 200 $\mu m$, indicating that our particles likely experienced a significant reduction of the impact angle and thus force relative to the impacts

analysed based on purely geometrically calculated impact angles. We conclude that these difficult to quantify first particle impacts into the curved side walls after a straight test section introduce some uncertainty but result in similar or in the worst case slightly higher impact forces compared to snow surface impacts. Based on the above discussion, we assume that the mechanical stresses affecting the snow particles in the curved section are comparable to real natural snow transport situations. A more in-depth analysis of the wall-impact forces would require detailed simulations or particle tracking measurements, which

is beyond the scope of this work.

### 3.4.2   Comparison to models

Our data facilitates a direct comparison to the predictions from SNOWPACK (Eq. 1, Lehning et al., 2002), SnowTran-3D (Eq. 2, Liston et al., 2007) and CROCUS (Eq. 3, Vionnet et al., 2012). All three models (Fig. 8) predict an increase of the densification rate with wind speed, however, significantly differ from each other and from our measurements when the original

parameters are used. Our measured densification rates fall right in between the range of values predicted by the models.

     All models are parameterized from field measurements, where simultaneous flow and microstructure measurements in unstable wind conditions with intermittent snow transport are difficult. The SNOWPACK model was parameterized based on data collected at an alpine catchment close to Davos, Switzerland (Weissfluhjoch, 2544m altitude). The density of the topmost 3 cm

of new snow were regularly sampled during different snowfall events, resulting in a temporal resolution of one sample per 0.5 - 1 h depending on the precipitation intensity. This is comparable to the timescale of our experiments with $\tau_{exp}$ = 0.5 h (Fig. 8). The SnowTran-3D model was parameterized based on data collected at rather smooth hills at a different climate in the US. We did not find any information on the timescales of the data set used for the SnowTran-3D parameterization (Liston et al., 2002). However, the higher densification rates predicted by the SnowTran-3D model (Fig. 8) may result from different time scales, atmospheric conditions, or additional transport in the absence of precipitation involved in the measurements they used for their parameterization. The parameters in Eq. (3) for the CROCUS model originate from a study carried out by Pahaut (1976) at Col de Porte (1325m altitude, French Alps). Unfortunately, no information on these measurements could be found.

We conclude that the differences between the models and our measurements are mainly the result of the estimated time scale ($\Delta t$) used for the calculation of the densification rates (Fig. 8). The new snow densification parameterizations (Eq. 1-3) do not contain any temporal component at all, although the measurements they are based on involved some time scales. However, densification of new snow under wind during precipitation events not only depends on the wind speed, but also on an effective transport duration ($\tau_t$) of individual precipitation particles, which is mainly governed by the precipitation intensity and particle cohesion as discussed below. We used a time scale of $\Delta t$ = 0.5h for calculating the densification rates for our experiments and all three models (Fig. 8). This time scale is at least appropriate for the SNOWPACK model and our measurements. That the SNOWPACK model nevertheless predicts significantly lower densification rates might be the result of lower precipitation rates during their field measurements resulting in longer effective transport durations $\tau_t$ as discussed in the following Section (Fig. 9a). The discrepancy for the two other models (SnowTran-3D and CROCUS) is likely also the the result of different time scales $\Delta t$ involved in their measurements used for the model parameterization. Changing $\Delta t$ from 0.5 h to 1 h for the SnowTran-3D model and to 0.1h for SNOWPACK and CROCUS already results in reasonable agreement of the models with our measurements, highlighting the strong dependency of the model on involved time scales. Additional discrepancies between the model descriptions and our measurements may arise from the fact that we did not consider additional compaction of surface snow layers due to wind when using the models (Fig. 8), because our RWT simulations are similar to the field measurements used to parameterize the wind speed dependent new snow density in the models. This highlights the problem of overlapping processes, where wind compaction during precipitation may be treated twice in the models: Once within the description of the wind speed dependent new snow density (Eq. 1-3), and once during additional wind compaction of surface layers. We conclude that a clearer separation in snowpack schemes may improve future model attempts of wind induced snow compaction, where the snowfall density only depends on temperature and humidity (to indirectly represent the variability in falling hydrometeors) and all the wind-related processes are treated by a well calibrated wind-compaction routine. Overall, the discrepancies between the models and our measurements can be attributed to poorly defined time scales, different precipitation intensities, different initial precipitation particles, particle cohesion, and local topography and climate conditions. This highlights the importance for more detailed physical descriptions of snow densification.

For completeness, we re-parameterize all three models by fitting them to our RWT data (Exp. 1-7, Table 1). We obtain the following parameters, $\beta_0$ = 2.16, $\beta_1$ = 0.034, $\beta_2$ = -0.63 and $\beta_3$ = 0.97 for the SNOWPACK (Lehning et al., 2002) model (Eq. 1) using the default 'ZWART' parameterization (Zwart, 2007), while $D_1$ = 17, $D_2$ = 250 and $D_3$ = 0.06 are found for the

SnowTran-3D (Liston et al., 2007) model (Eq. 2). The following parameters $a_\rho$ = 43 kg m$^{-3}$, $b_\rho$ = 9 kg m$^{-3}$ K$^{-1}$ and $c_\rho$ =

35 kg m$^{-7/2}$ s$^{-1/2}$ were obtained for the CROCUS model when fitting Eq. 3 to our data. Minimum air temperatures of $T_a$ = -10°C (CROCUS) and $T_a$ = -15°C (SnowTran-3D) had to be used for these model fits instead of the actual air temperature $T_a$ = -24°C measured during the Experiments 1-7 (Table 1), because the models result in unrealistic values (negative densification rates) for lower air temperatures. However, Fig. 9b in our manuscript shows that the densification rate tends to be temperature independent below approximately $T_a < -6°$. Therefore, the fit parameters for the crocus model are only considered to be valid

for $V_{0.4m} > 3.8$ m $s^{-1}$ in Eq. 3, which corresponds to a wind speed of 5 m $s^{-1}$ at a height of 2m with an aerodynamic roughness length of $z_0 = 0.24$ mm for fresh snow as determined by Gromke et al. (2011).

The current separation into precipitation and no-precipitation events in the three models as discussed in Section 2.4 results in a gray zone where processes may overlap. New, temporally highly resolved models may aim for more physically based descriptions of these processes. Therefore, a particle shape-based parameterization of the density and SSA as proposed in

CROCUS (Vionnet et al., 2012) based on dendricity and sphericity in combination with an effective particle transport duration $\tau_t$ would likely be beneficial to simultaneously cover precipitation and no-precipitation events in future modeling attempts. However, these parameters are very difficult to measure and quantify experimentally, which is probably the reason why current snow models prefer using simple empirical correlations instead of physically based process descriptions.

### 3.4.3 Sensitivity to duration and temperature

The measurements with different experiment durations (Experiment 8-9, $\tau_{exp}$ = 1 h and 2.5 h, Table 1) and air temperatures (Experiment 10-12, $T_a$ = -12°C, -6°C and -2°C, Table 1) were conducted at a wind speed of $V_{0.4m}$ = 6 m s$^{-1}$ and are spanning a range of densification rates from 1.5 h$^{-1} < \phi_{rate} < 4$ h$^{-1}$ (Fig. 9). An effective transport duration $\tau_t$ can be interpreted as the average time individual snow particles are effectively transported by the wind, whereas the experiment duration $\tau_{exp}$ (Table 1) is defined as the time during which one box of new snow is slowly added to the wind tunnel simulating precipitation. Therefore,

an increasing $\tau_{exp}$ can thus be interpreted as a decreasing precipitation rate. The transport duration $\tau_t$ is difficult to quantify for these experiments. However, $\tau_t$ is a function of $\tau_{exp}$, so $\tau_t$ can be indirectly controlled by varying the time a box of snow is slowly added to the RWT and thus by $\tau_{exp}$. By reducing this 'precipitation rate' for the experiments 8 ($\tau_{exp}$ = 1 h) and 9 ($\tau_{exp}$ = 2.5 h), the effective transport duration $\tau_t$ of individual snow grains is increased as they are longer exposed to the wind before ultimately deposited and covered by additional new snow, which was argued to be an effect of additional fragmentation.

The decrease of the densification rate with experiment duration (Fig. 9a) would be consistent with this: Fragmentation should be high initially and decreasing with time, since the most fragile particles break first. If thus fragmentation controls densification, a decrease of the rate as a function of transport duration can be expected as observed in Fig 9a. A simple two parameter fit of a function inverse proportional to the experiment duration (similar to the definition of the $\phi_{rate}$) is applied to the data in Fig. 9a resulting in

$\phi_{rate}(\tau_{exp}) = A_1/\tau_{exp} + B_1$ (5)

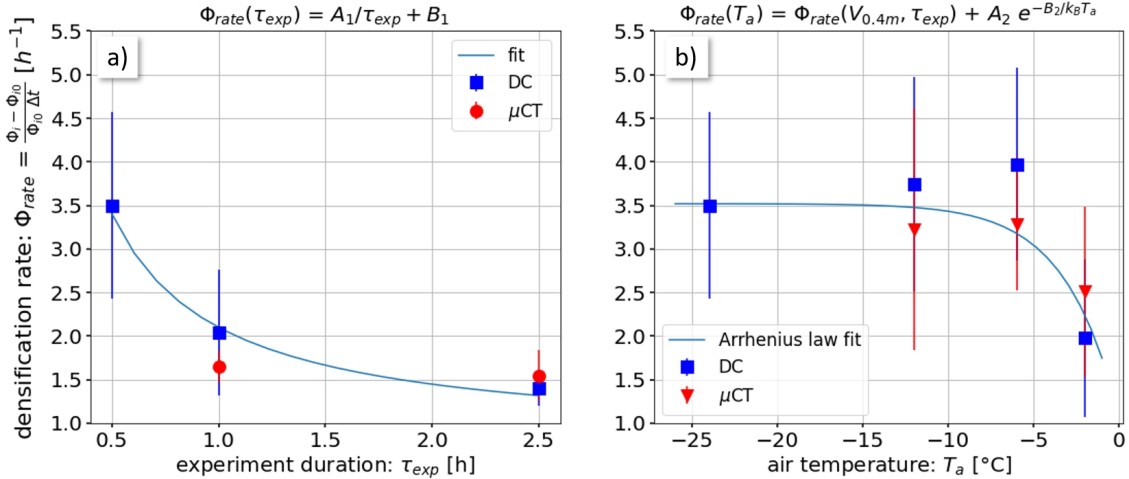

**Figure 9.** Densification rate as a function of a) the experiment duration $\tau_{exp}$ and b) the air temperature $T_a$ including fits.

with $A_1 = 1.30$ and $B_1 = 0.80$ h$^{-1}$, simply to represent the three data points in the experimental range with a two-parameter fit. On physical grounds, the densification rate should tend to zero instead for very long times which would require more data points to obtain a reasonable fit. However, the good fit of the reciprocal function (Eq. 5) indicates that the time $\tau_{exp}$ (experiment duration) governs the decrease of the densification rate and not the change in ice volume fraction.

The temperature dependence of the densification rate shows little or no trend for temperatures below $T_a < $ -5°C (Fig. 9b) with values of $\phi_{rate} \approx 3.5$ h$^{-1}$, while it significantly drops to $\phi_{rate} \approx 2.2$ h$^{-1}$ for higher air temperatures $T_a > $ -5°C (Experiment 12). Cohesion of the snow particles drastically increases for $T_a > $ -5°C as shown by the angle of repose experiments performed by Willibald et al. (2020) and Eidevåg et al. (2022). We hypothesize that cohesion and sintering at higher air temperatures inhibits snow particle transport, saltation and thus fragmentation and snow densification. This hypothesis cannot be proven based on our RWT experiments presented in this study. However, for another RWT study, we simulated precipitation while continuously increasing the air temperature from $T_a = $ -5°C to +2°C. We found that the particle mass flux in the saltation layer gradually decreased and entirely stopped at around -0.5°C to 0°C due to strong cohesion between the new snow crystals on the ground resulting in low density snow accumulations. As in Willibald et al. (2020), we can assume an Arrhenius form for the temperature dependency of the densification rate:

$\phi_{rate}(T_a) = \phi_{rate}(V_{0.4m}, \tau_{exp}) + A_2 e^{-B_2/k_B T_a}$          (6)

where $\phi_{rate}(V_{0.4m} = 6$ ms$^{-1}$, $\tau_{exp} = 0.5$h$) = 3.5$ h$^{-1}$ is the mean initial densification rate for $T_a < $ -5°C, $A_2 = $ -3.12 $10^{38}$ h$^{-1}$ and $B_2 = 2.06$ eV are constants, and $k_B = 8.6 \ 10^{-5}$ eVK$^{-1}$ the Boltzmann constant, thus the air temperature in Kelvin must be used in Eq. 6. While the inferred parameterization $\phi_{rate} = f(T_a)$ is based on limited data and a limited range of parameter validity, the data highlights a similar temperature effect in wind-deposited snow observed here and ballistically deposited snow
governing the angle of repose (Willibald et al. 2020).

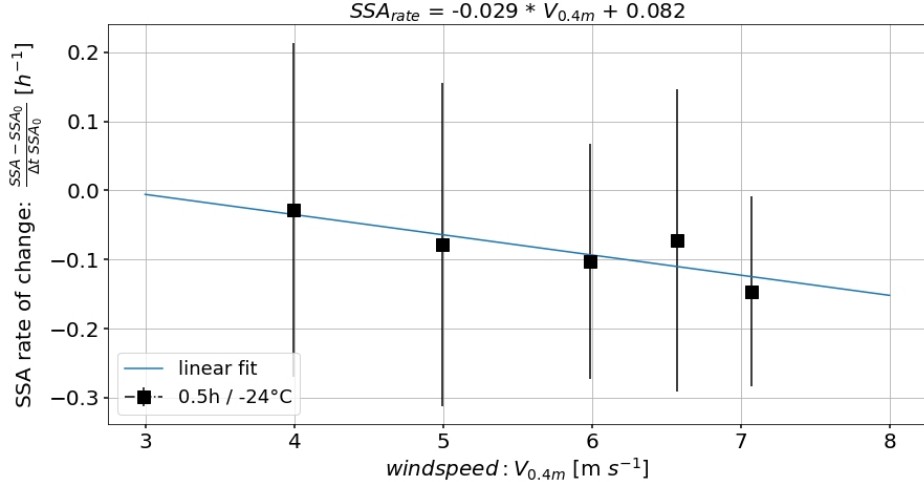

**Figure 10.** $SSA_{rate}$ as a function of wind speed $V_{0.4m}$ for the experiments 3-7 (Table 1) including a linear fit.

### 3.5 $SSA$ rate

The impact of wind speed, transport duration and air temperature on the $SSA$ rate, defined as $SSA_{rate}$ = ($SSA$- $SSA_0$) / ($SSA_0$ $\Delta t$) is discussed in this Section in analogy to the densification rate.

#### 3.5.1 Dependence on wind speed and involved mechanisms

Fig. 10 shows that the $SSA_{rate}$ increases in absolute values from $SSA_{rate}$ = -0.03 h$^{-1}$ to $SSA_{rate}$ = -0.16 h$^{-1}$ with increasing wind speed ($V_{0.4m}$ = 4 - 7.1 ms$^{-1}$). Although the $SSA$ decrease is generally small relative to the error bars (Fig. 6c-d, partially due to the spatial $SSA$ variability, Fig. 7b), the data point for $V_{0.4m}$ = 7.1 ms$^{-1}$ confirms a significant deviation from $SSA_{rate}$ = 0 h$^{-1}$. These rates are up to one order of magnitude higher than the rates ($SSA_{rate} \approx$ -0.01 h$^{-1}$) found by Schleef et al. (2014a) for isothermal metamorphism. Our result substantiates the findings of Cabanes et al. (2003) who found wind to 465 increase the rate of $SSA$ decrease which was not further quantified in their study due to limited data.

Fragmentation of dendritic crystals cannot cause the measured reduction in the $SSA$, because the resulting additional surfaces would lead in an increase in the ice surface and thus the $SSA$. Different mechanisms remain that may be responsible for the $SSA$ decrease: i) Sublimation: Some of the smallest fragments and dendrites entirely disappear because of sublimation (Groot-Zwaaftink et al., 2013). This assumption is supported by the increase in relative humidity as shown in Fig. 2. ii) 470 Vapor re-deposition: Recrystallisation of humidity at the surface of suspended snow particles likely results in an average grain growth similar to isothermal metamorphism of new snow (e.g., Schleef et al., 2014a). These first two mechanisms we refer to as '$airborne\ snow\ metamorphism$', in analogy to sublimation and re-deposition processes occurring in isothermal deposited snow. iii) Particle separation: This effect has not yet been documented for snow, however, likely occurs when larger fragments remain in saltation close to the snow surface (Fig. 3) whereas the smallest fragments transition into suspension (Nishimura

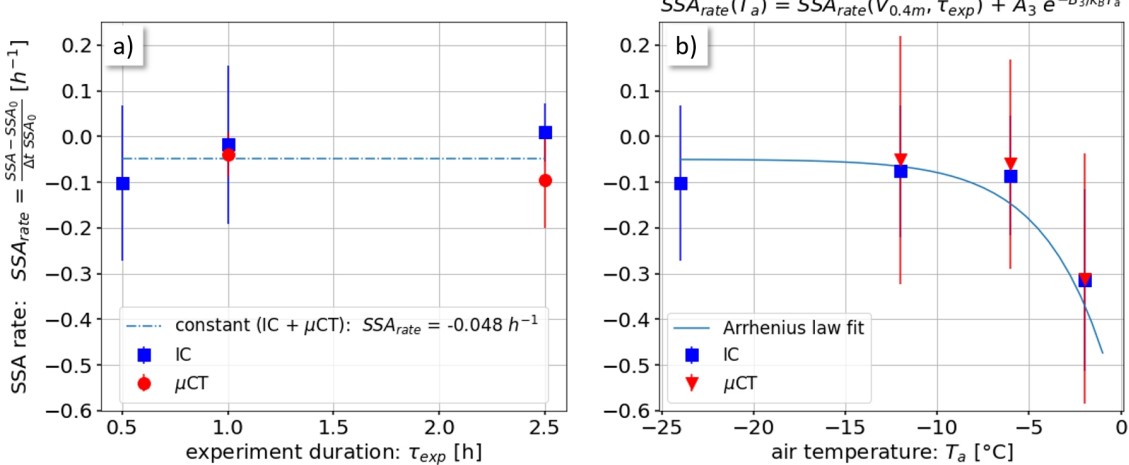

**Figure 11.** $SSA_{rate}$ as a function of a) the experiment duration $\tau_{exp}$ and b) the air temperature $T_a$ including an exponential fit.

and Nemoto, 2005), forming a blowing snow layer (Palm et al., 2017). In the field, these small blowing snow particles may disappear due to sublimation or being blown out of an area of interest (Palm et al., 2017). In a non-sealed RWT, small particles may be also blown out at the inlet (Fig. 1a). Thin layers of fine snow dust were found after the experiments outside the RWT in proximity to the inlet (Fig. 1a) supporting that particle separation occurs in our experiments.

The minor, potentially linear increase of the negative $SSA$ rates (in absolute values) in Fig. 10 suggests an amplification
of the above assumed processes of airborne metamorphism and particle separation with increasing wind speed. We argue that higher airborne particle concentrations at higher wind speeds allow an increasing number of particles experiencing airborne metamorphism. Higher mass fluxes also result in an increasing number of smaller particles in higher air layers (Walter et al., 2017) that may partially be blown out of the inlet, thus likely resulting in intensified particle separation. These processes are further analyzed based on additional sensitivity experiments introduced and discussed in Section 3.6.

**3.5.2 Sensitivity to duration and temperature**

The $SSA_{rate}$ for the different experiment durations (Experiment 8-9, $\tau_{exp}$ = 1 h and 2.5 h) and air temperatures (Experiment 10-12, $T_a$ = -12°C, -6°C and -2°C) span a range of -0.3 h$^{-1}$ < $SSA_{rate}$ < 0.01 h$^{-1}$ (Fig. 11). Regarding the dependency of $SSA_{rate}$ on the experiment duration $\tau_{exp}$ (Fig. 11a) a mean value of $SSA_{rate}$ = -0.048 h$^{-1}$ is found. Due to the limited data and the relatively large error bars, no firm conclusion can be drawn on the dependency of $SSA_{rate}$ on the experiment
duration $\tau_{exp}$ based on Fig. 11a. However, a supplement experiment introduced in Section 3.6.2 demonstrates a distinct effect of airborne snow metamorphism on the $SSA$ decrease and its dependency on the effective transport duration $\tau_t$.

A different picture appears for the dependency of the $SSA_{rate}$ on the air temperature $T_a$ (Fig. 11b), where a constant value of $SSA_{rate}$ = -0.07 h$^{-1}$ is measured for $T_a$ < -5°C, while a significant drop to a value of $SSA_{rate}$ = -0.32 h$^{-1}$ is measured for Experiment 12 at $T_a$ = -2°C. Despite the rather large error bars (see Section 3.3), the significant change in $SSA_{rate}$ for

$T_a$ > -5°C has individually been measured by two independent instruments, with good agreement between the $SSA_{rate}$ values determined from the IceCube (blue squares) and the µCT (red triangles) measurements. While stronger cohesion was likely the reason for the reduced densification rate $\phi_{rate}$ for $T_a$ > -5°C (Fig. 9b), enhanced sublimation (Palm et al., 2017) and vapor re-deposition at higher air temperatures may explain intensified airborne snow metamorphism and thus a stronger reduction of the $SSA$ values in this case (e.g. Harris et al., 2023). Like for the densification rate (Eq. 6), we assume an Arrhenius form

(Willibald et al., 2020) for the temperature dependency of the $SSA$ rate:

$$SSA_{rate}(T_a) = SSA_{rate}(V_{0.4m}, \tau_{exp}) + A_3 e^{-B_3/k_B T_a} \tag{7}$$

where $SSA_{rate}(V_{0.4m} = 6 \text{ m s}^{-1}, \tau_{exp} = 0.5h) = -0.07 \text{ h}^{-1}$ is the mean initial $SSA_{rate}$ for $T_a$ < -5°C, $A_3 = -5.4\ 10^{33} \text{ h}^{-1}$ and $B_3 = 1.8$ eV are constants, and the air temperature $T_a$ in Kelvin must be used in Eq. 7. Whether the proposed parameterization of Eq. 7 is valid for different wind speeds $V_{0.4m}$, experiment durations $\tau_{exp}$ and relative humidity $RH$ must be tested in future

studies. Similar results of a decreasing $SSA$ with increasing wet-bulb air temperature were found by Yamaguchi et al. (2019) for precipitation particles under low wind conditions, while also Schleef et al. (2014) found that the $SSA$ decay increased with higher air temperatures for isothermal snow metamorphism.

### 3.6    Particle separation and airborne snow metamorphism

To further substantiate the previous hypotheses that airborne snow metamorphism and particle separation play a role for the

evolution of the $SSA$ under the influence of wind we provide sensitivity evaluations.

### 3.6.1    Particle separation

For Experiment 9 we also collected some of the accumulated dust outside of the RWT for µCT analysis, in addition to the initial snow and the snow deposited in the RWT. A metric for the particle size distribution can be derived from the µCT images by filling the 3D ice matrix with inscribed spheres of different diameters $D$ (Hildebrand and Rüegsegger, 1997). Figure 12a shows

the frequency distribution of the spheres for the three snow samples taken during Experiment 9 (Table 1). Similar distributions were obtained for all experiments (5-12) where µCT samples were taken. The dendritic new snow produced in the Snowmaker (cyan squares) shows a peak in the distribution at a size of $D_{peak} = 80$ µm and only few particles at a maximum size of $D_{max} = 300$ µm. A general shift of the distribution towards larger sphere sizes ($D_{peak} = 100$ µm), along with a reduction in the frequency of smaller spheres (20 µm < D < 80 µm) and a maximum size increase to $D_{max} = 900$ µm is found for the

accumulated snow after wind exposure (green dots). This shift indicates that smaller particles, e.g. fragmented dendrites are removed from the system likely due to sublimation and particle separation where, for the latter, the smallest fragments are partially blown out of the RWT. The increased appearance of particles with D > 300µm after wind exposure may also be the result of particle separation, where the largest particles remain in saltation or roll on the ground and are finally deposited in the accumulation area. The general shift of the particle size distribution towards larger spheres due to particle separation and

airborne metamorphism is in line with our finding of a decreasing $SSA$ as discussed in the previous Section.

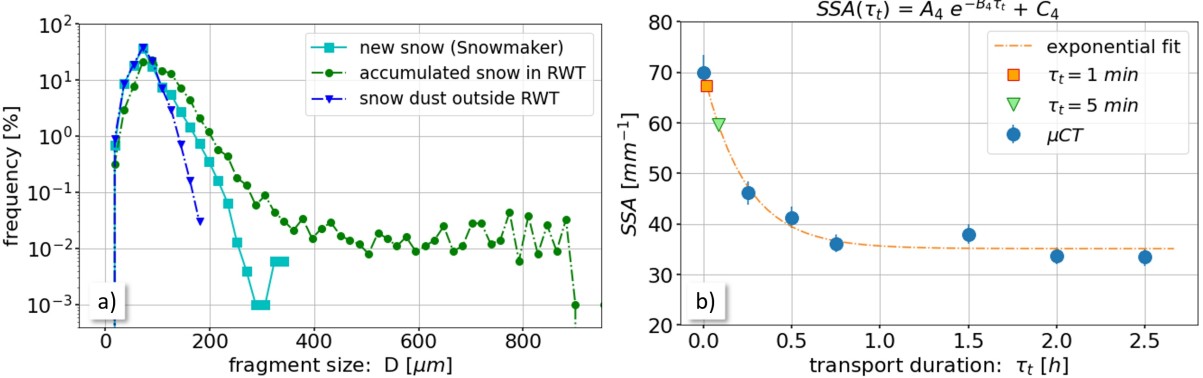

**Figure 12.** a) Particle size distributions derived from $\mu$CT measurements for the fresh snow produced in the Snowmaker (cyan squares), the wind affected snow sampled in the accumulation area (green dots), and for snow dust that has been blown out of the RWT for experiment 9 (blue triangles). b) Exponential $SSA$ decrease measured for the supplementary experiment on long transport durations $\tau_t$ including estimates for the effective transport durations of the main experiments (orange square and green triangle).

The existence of the process of particle separation in our RWT experiments is substantiated by the particle size distribution of a small snow dust sample taken outside of the RWT (blue triangles, Fig. 12a). Most of these snow dust particles are within a range of $20\mu$m < D < $120\mu$m and coincide exactly with the size range where a depletion is found for the wind affected snow (green circles) relative to the initial new snow (cyan squares). Generally, particle sizes in snow saltation layers

were experimentally (Gromke et al., 2014) and numerically (Melo et al., 2021) found to strongly decrease with increasing height above the ground, being the reason that only the smallest particles are blown out at the inlet. Similar processes of particle separation occur during Saharan dust events, where micrometer sized particles are transported hundreds or thousands of kilometers whereas larger sand grains are predominantly redeposited locally (e.g. van der Does et al., 2016). For ice deserts, sublimation of blowing snow particles is resulting in a significant snow mass loss for Antarctica (Palm et al. 2017). Apparent

similarities are also found between our wind affected particle size distributions and the model results from Comola et al. (2017). While this analysis of size distributions is consistent with particle separation and the hypothesized impact on $SSA$ decay, we acknowledge the known difficulties of the involved size distribution (Löwe et al., 2011) for characterizing the full size complexity of dendritic crystals. A further analysis likely requires a better metric to discern fragmentation and separation.

### 3.6.2 Airborne snow metamorphism

In our main experiments, the $SSA$ decay in the deposit (Fig. 6c-d, 10 and 11) is only moderate and the hypothesized process of airborne snow metamorphism through continuous sublimation and vapor re-deposition is difficult to verify. To illustrate the existence of airborne metamorphism and its relevance for the $SSA$ decrease, a sensitivity RWT experiment was conducted where a single portion (600g) of dendritic new snow from the snowmaker was added to the initially empty RWT at $T_{air}$ = -20°C, while the wind speed was set to the maximum of $V_{0.4m} = 8\ ms^{-1}$ (Experiment 14). In this way we were able to maintain

a large portion of the particles in suspension over long periods without deposition. Snow was periodically collected from the suspended particle phase by capturing particles with a small bag for $\mu$CT analysis out of the air stream. For this experiment, the RWT was sealed as good as possible to avoid particles being blown out of the RWT. A more detailed description of this additional experiment and profound evidence of the existence of airborne snow metamorphism based on stable water isotope analyses can be found in Wahl et al. (2024).

Fig. 12b shows a considerable decrease of the $SSA$ with increasing transport duration $\tau_t$. Here, the real transport duration for the particles is used instead of the previously (Fig. 9a and 11a) used experiment duration $\tau_{exp}$, as it can be assumed that the particles that are captured out of the flow after some time were mobile the majority of the preceding time. The decrease can be empirically described by an exponential function:

$$SSA(\tau_t) = A_4 e^{-B_4 \tau_t} + C_4 \tag{8}$$

where $A_4 = 34.5$ mm$^{-1}$, $B_4 = 4.2$h and $C_4 = 35.2$ mm$^{-1}$ are constants. The observed change of the snow microstructure from highly dendritic crystals to continuously growing rounded grains together with the strong decrease in the $SSA$ demonstrates the existence of airborne snow metamorphism, with an impact on the $SSA$ decrease which depends on transport duration.

We can compare our previous results with Figure 12b by the following reasoning. The average transport duration $\tau_t$ for the main RWT experiments can be roughly estimated when dividing the experiment duration ($\tau_{exp}$) (Table 1) by the number of snow supplies ($\approx$30) when simulating precipitation. This leads to an estimate of $\tau_t \approx 1$ min for the 0.5 h lasting experiments and $\tau_t \approx 5$ min for the 2.5 h lasting experiment. The corresponding $SSA$ reductions are shown in Fig. 12b, implying a reduction of the specific surface area from $SSA = 70$ mm$^{-1}$ to $SSA = 67$ mm$^{-1}$ for the 0.5 h experiments (orange square), and a reduction to $SSA = 60$ mm$^{-1}$ for the 2.5 h experiment (green triangle in Fig. 12b). The corresponding $SSA$ rates are in both cases $\phi_{rate} \approx -0.1 h^{-1}$ which is consistent with the rates measured for the main RWT experiments (Fig. 10 and Fig. 11). We thus expect these rates to be a real signature of airborne snow metamorphism, that is concealed in the uncertainties due to the short transport durations, spatial variability and measurement uncertainties in our experiments.

## 4  Conclusions

The focus of this study is on linking atmospheric and aeolian snow transport conditions during precipitation events to the snow microstructure of the ultimately deposited snow and to identify the relevant processes. Therefore, ring wind-tunnel experiments were performed to investigate the evolution of the surface snow microstructure (specific surface area and density) under wind for different transport conditions (wind speed, air temperature and transport duration) in a controlled cold laboratory environment. The results provide novel insight into the link between atmospheric conditions, airborne snow particles and deposited snow. While we show magnitudes of dependencies between different flow and snow parameters, processes and snow microstructures, natural conditions may be different in the field, depending on the snow type or flow conditions, while the latter is also rarely well developed and stationary for natural conditions.

The measured increase of the densification rate with increasing wind speed (Fig. 8) significantly differs from the increase of previous model parameterizations that are exclusively based on field studies. Therefore, a re-parameterization of these models from our data is derived. The dissipated energy upon particle impact (Fig. 5) suggests that enhanced fragmentation of dendritic snow crystals along with higher packing densities is the reason for increasing snow densities at higher wind speeds. A decreasing densification rate with increasing transport duration (Fig. 9a) is consistent with that, since the initial fragmentation must be higher where the most fragile dendrites or particles break first. For higher air temperatures ($T_a$ > -5°C), the densification rate show a marked decrease compared to the rather constant rates at lower temperatures (Fig. 9b). This was attributed to the effect of enhanced cohesion (or sintering) which on one hand inhibits particle transport and fragmentation and thus reduces the packing fraction directly close to the melting point.

The observed slight enhancement in $SSA_{rate}$ (Fig. 10) with increasing wind velocity is attributed to the processes of airborne snow metamorphism and particle separation. Both are expected to be amplified with increasing wind velocities. Similar to the densification rate, the $SSA$ decrease rate was approximately constant at low air temperatures ($T_a$ < -5°C) and markedly increased (in absolute values) for air temperatures ($T_a$ > -5°C). The significance of airborne snow metamorphism, where sublimation and vapor re-deposition result in a $SSA$ reduction, was demonstrated in a sensitivity experiment which revealed a strong $SSA$ decrease of the airborne particles for long transport durations (Fig. 12b). The process of particle separation, where some of the smallest particles in suspension are blown out of the wind tunnel, was exemplified by an analysis of the particle size distribution of a small snow dust sample taken outside from the wind tunnel roof; however, its relevance for real natural blowing snow events remains debatable. Although the processes of airborne snow metamorphism and particle separation are certainly not perfectly reproduced in the RWT relative to real field conditions due to the uncontrolled exchange of air with the outside of the RWT at the inlet, they both occur during natural aeolian snow transport. Therefore, the observed $SSA$ decrease (Fig. 10, 11 and 12b) provides a first assessment of the impact of these processes on the $SSA$, however, the magnitudes of the $SSA$ decrease must be verified in future studies.

Overall, we have demonstrated that our setup of a ring wind-tunnel in a cold laboratory allows for revealing the relevant processes responsible for snow densification and $SSA$ decrease under wind. The presented results highlight the complexity of the involved processes and how they may affect the resulting surface snow microstructure. However, more detailed studies on the individual processes, i.e. particle fragmentation, airborne snow metamorphism and particle separation are required to fully understand and quantify their impact. Our study also emphasizes the importance of developing process based rather than empirically based parameterizations for the evolution of the surface snow microstructure under wind. Such improved model descriptions of surface snow will especially be relevant for applications in radiative transfer (snow albedo or radar remote sensing), where the $SSA$ is the dominant parameter. Furthermore, the exchange of chemical species with the atmosphere, snow hydrology or avalanche prediction will also profit from a better understanding and prediction of the surface snow microstructure. Finally, any layer within a snowpack originates from a surface snow layer. A detailed understanding of the initial microstructure of the surface snow layer is therefore critical for the evolution of any layer within the snowpack throughout its (seasonal) lifetime.

*Data availability.* The data will be made available after the manuscript is officially published.

*Author contributions.* BW, HW and SW conducted the experiments, BW analysed the data, created the figures, and prepared the manuscript. HW analysed the high-speed camera data. HL provided theoretical input and edited the manuscript.

*Competing interests.* The authors declare no competing interests.

*Acknowledgements.* The authors acknowledge Hongxiang Yu for providing OpenFoam simulations of the air flow and the particle saltation layer in the ring wind tunnel, Matthias Jaggi for the cold laboratory support, and Christian Sommer and Charles Fierz for the design and realization of the ring wind tunnel system.

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
