# Peer review of "Wind tunnel experiments to quantify the effect of aeolian snow transport on the surface snow microstructure"

_The Cryosphere, 2023_

## Author Comment (AC1)

**RC1**: 'Comment on tc-2023-112', Anonymous Referee #1, 15 Sep 2023

The authors present a very exciting and compelling experiment focused on the effect of wind snow surface microstructure. For how simple the question is, this is an incredibly hard problem to work on. We have limited tools at our disposal to make concise measurements of snow microstructure, and it is incredibly difficult to run such an experiment in the field. This group at SLF has succeeded at combining their expertise in snow microstructure and wind tunnel experiments to provide new insights into this intriguing aspect of snow metamorphism.

*First thanks a lot for the appreciation of our study and the detailed review of the manuscript. We included basically all your suggestions which were clear and concise, and which helped to significantly improve the article!*

My only concern with this research is the significantly unphysical conditions under which snow is being transported. As it stands, I do not see a reason why the rate of change of any snow characteristics in their experiment should be related to any measurements of natural snow undergoing natural transport.

*Thank you for this important comment. The particle impact characteristics shown in Fig. 4 indicate a sufficiently developed, physical reproduction of a well-developed snow saltation layer similar to natural conditions in the straight sections of the ring wind tunnel (RWT). In the curved sections, impact forces are comparably small due to the relatively large radius and the fact that the particles are sliding along the wall instead of impacting as we discuss below.*

*The focus of our study is on linking atmospheric and aeolian snow transport processes to the snow microstructure of the ultimately deposited snow. While we show magnitudes of dependencies between different flow and snow parameters, processes and snow microstructures, natural conditions may be different in the field, depending on the snow type or flow conditions, while the latter is also rarely well developed and stationary for natural conditions. Our reparameterization of existing models aims at providing a connection between our lab results and field parameterization, while our results also show that new more process based parameterizations are required to better represent the effect of air temperature, wind speed or precipitation intensity.*

It is unclear if the snow particles actually come in contact with the propeller driving their RWT. A schematic that shows this mechanism would be very helpful.

*The propeller is located beneath the lid in the curved section of the ring wind tunnel (Fig. 1a) covering approximately the top quarter of the wind tunnel cross-section. The propellor blades are 90 mm long. As the particle mass flux exponentially decreases with height as shown by Yu et al. (2023) for our RWT, only a negligible part of the snow particles will get into contact with the propellor. A schematic drawing of the RWT and additional figures can be found in Yu et al. (2023) and will be referenced in the revised version.*

More importantly, the authors acknowledge that a large portion of snow particles are transported along the outer wall due to centrifugal forces and, among other effects, this causes a measurable impact on density. This is well outside the realm of normal saltation and suspension. Given that $v_x$ is so much larger than $v_z$, this impact force may be considerably higher than in nature. As

well, repeat impacts caused by snow working its way around a corner may cause orders of magnitude more fragmentation.

*In the curved sections, particles are mainly sliding along the RWT outer wall, an effect that is certainly not favourable for simulating natural snow transport but inevitable for a compact closed circuit wind tunnel in a cold laboratory of limited dimensions. The centrifugal forces acting on snow particles in the curved section were estimated being two to three orders of magnitude smaller compared to the forces acting on the snow particles during surface impact while saltating. The maximum impact angles of snow particles impacting into the curved outer wall were calculated being around 25°-30° which are comparable to the impact angles $\alpha_{in}$ on the snow surface in the straight test section (Fig. 4b and d). However, a Stokes number < 0.1 indicates a good flow following behaviour of the snow particles when the air flow gets redirected in the curved section, resulting in actually much smaller impact angles. Based on these results, we conclude that both effects have a similar or smaller effect on particle fragmentation than particles impacting on the surface during saltation. We will add this discussion in the revised version in Section 3.2.*

Given these concerns, could you please address the question of transport around the curves (impact velocities, momentum balance, fragmentation rate, restitution coeff, how many more impacts per second? etc.), or modify the manuscript in such a way that the reader knows while you may have novel measurements of a physical process, this physical process has little relation to what one may expect to find in nature? As it stands, I think the quantitative information provided needs to be qualified or better justified.

*Based on the above estimates on centrifugal forces and impact angles it can be assumed that particle fragmentation is dominated by the particle impacts in the saltation layer in the straight sections of the RWT, and that our results are, to a certain degree, comparable to real natural snow transport situations.*

There are a few grammatical things that could be improved:

L8- Cover wind speeds? *We change this to: "vary wind speeds".*

L11-In the deposit? *We change this to: "in the deposited snow".*

L20- Is rolling different from creep? *No: we will call it "rolling or creeping"*

L29-Chemical species? *We change this to "Chemical substances".*

L60: Do you mean necessary or inevitable? *We change this to "necessary".*

L161: To make contact to previous studies? *We change this to: "To compare the quality of our particle transport phase to previous studies".*

Other comments

L53-54- At what height are these wind velocities? *We add here the height of 1 m.*

L76-77: Very cool Thanks.

L80-82: Do the particles not come in contact with the propeller? *Only a negligible number of particles gets into contact. The majority of the particles is transported close to the ground in the lowest 5-10 cm. Please also see our response to your previous comment above.*

Figure 2: How did you conclude the jump in RH was from snow particle sublimation? What's the RH of the cold room? *The relative humidity of the cold room varies depending on how often people enter and leave the room. In the morning, the RH is typically low at around 40% - 50%. Therefore, depending on the initial RH before an experiment, a mor or less strong jump in RH is obtained due to sublimation of the suspended particles (Dai and Huang, 2014).*

L258-259: Very cool *Thanks.*

L267-268: Again, how can you decouple this from the effect of particles smashing into walls that are necessarily there in nature? *As discussed above, the forces acting on the particles in the curved section due to centrifugal effects and while impacting on the side walls are estimated being similar at most or smaller than the forces they experience during saltation.*

---

## Author Comment (AC2)

**RC2**: 'Comment on tc-2023-112', Anonymous Referee #2, 15 Sep 2023

**Review of the paper "Wind tunnel experiments to quantify the effect of aeolian snow transport on the surface snow microstructure" by Walter et al. submitted to The Cryosphere.**

This paper presents an innovative set of measurements to investigate the effect of wind-induced snow transport on the physical properties of surface snow (density, SSA). These measurements were collected in a ring-shaped wind tunnel (RWT) that reproduces the main characteristics of aeolian snow transport. The authors quantified the changes in density and SSA during events with different wind speed and air temperature. Their analysis confirmed the increase in surface snow density with increasing wind speed that has been observed in the field and highlighted a slight decrease in SSA with increasing wind speed. The author also compared the densification rates measured in the RWT with parameterizations used in snowpack schemes.

The subject of this paper is very interesting for the snow community and presents a set of original measurements to quantify the effects of wind on the physical properties of surface snow. So far, these quantifications have mainly been obtained from field measurements (mainly for surface snow density) that are influenced by other physical processes, making it challenging to disentangle the effect of the wind from the other processes. These measurements can serve to develop more-physically based parameterizations of the impact of wind on the physical properties of the snow cover in multi-layer snowpack schemes such as Crocus and SNOWPACK. Therefore, this paper should be published in The Cryosphere. However, prior to publication, the author must carefully define in which context they are working (blowing snow event with concurrent snowfall) and revise accordingly which existing parameterizations they are evaluating in the context of the study. These two general comments are followed by more specific and technical comments.

*First thanks a lot for the very detailed review of our manuscript. We included most of your suggestions which helped to significantly improve the quality of the article!*

**General comments**

1. In this paper, the authors study the effects of aeolian snow transport on the properties of surface snow using a RWT where snow is continuously added to mimic snow precipitation until the end of the different experiments. In this respect, the authors are reproducing in their experiments what happened during blowing snow events with concurrent snowfall. In such conditions, most of the snow transported by the wind is made of precipitating snow particles that fall continuously on the snow surface and are then transported by the wind as illustrated nicely by Figure 3 in the submitted paper. Snow particles initially present at the snow surface may also be transported if the wind speed is sufficient. This situation differs from blowing snow events without snowfall when the transported snow is only made of snow initially present at the snow surface without the constant supply of new snow particles from snowfall. Even if the physical processes involved are the same in terms of particle impacts and ejections, we can expect different densification rates and changes in SSA due the influence of the constant supply of fresh dendritic snow during blowing snow events with concurrent snowfall. Therefore, I strongly

recommend to the authors to define well the type of blowing events that they are simulating in the wind tunnel and to discuss how the constant supply of fresh snow influences the results presented in their study. Without such clear discussion, their conclusions may be applied erroneously to different situations that were not captured yet in the RWT experiments.

*We agree that we need to better highlight (especially in the abstract and elsewhere) that we only consider blowing snow events with concurrent snowfall. Our supply of fresh snow simulating precipitation leads to a constant source of highly fragile dendritic new snow crystals that are primarily transported by the wind. Our results (Fig. 9a) shows that initial densification rates are highest which we attribute to the effect that the most fragile dendrites are easily fragmented once added to the flow. We will revise the manuscript to better point out that we only consider precipitation events.*

2. The authors have evaluated two parameterizations used in snowpack schemes for snow densification that account for the effects of wind. This part of the paper is very valuable for snowpack schemes, but it should be strongly revised to reflect well how the effects of wind on surface snow properties are included in snowpack schemes and then to make sure that the correct parameterizations are tested in this study.

The effect of wind in snowpack schemes such as Crocus, SNOWPACK and SNOWMODEL consist in a two-step process: (i) the falling snow density generally includes a dependency on wind speed (Pahaut, 1975; Lehning et al., 2002; Liston et al., 2007). It is computed at each model time step based on the meteorological forcing (wind speed, air temperature, …) and new snow is added at the top of the snowpack, (ii) the models then account for wind-driven compaction by including a wind compaction term when calculating compaction in the near surface snow layers (Brun et al., 1997; Liston et al., 2007; Amory et al., 2021; Wever et al., 2022). The wind compaction rate depends on the intensity of aeolian snow transport. This second component allows the models to simulate the increase in surface density during blowing snow events without snowfall (cf my first general comment). So far, in their paper, the authors are evaluating two parameterizations for the density of new snow and are ignoring the second component of wind-driven compaction in numerical snow models. I recommend them to better justify why they are only evaluating the first component. There is a clear grey zone in between these two model components with parameterizations that may overlap and may treat twice the same physical process. Ultimately, we could imagine a clearer separation in snowpack schemes where (i) the snowfall density only depends on temperature and density (to indirectly represent the variability in falling hydrometeors) and (ii) all the wind-related process are all treated by a wind-compaction routine. Under this assumption, the measurements presented in this paper would serve to adjust such wind-compaction routines.

*Thanks for this important comment! We agree that we need to better clarify why we chose these model components. From both models (Lehning et al., 2002 and Liston et al., 2007) we used the parameterizations (EQ. 1 and 2 in our manuscript) that define a new snow density for precipitation events under the influence of strong winds as we simulated in our case in the RWT. Both models are parameterized based on field measurements of new snow densities with concurrent snowfall. Therefore, we do not consider the effect of snow densification due to wind in the absence of precipitation.*

*The current separation into wind affected precipitation and snow transport without precipitation in models is likely resulting from different time scales and snow types involved. During precipitation events, the typically highly dendritic new snow will quite quickly (depending also on the wind speed) cover the underlying (new) snow which then can't be entrained anymore by the wind, resulting in rather short transport durations $\tau_t$ as discussed in our Section 3.6.2. However, without precipitation, (loose) surface snow of likely different grain type (potentially more decomposed rounded particles) may be entrained and affect by wind for much longer time (or transport) durations.*

*Regarding the transport duration: Fig. 9a shows the dependency of the densification rate for different experiment durations ($\tau_{exp}$) with different effective transport durations ($\tau_t$) as estimated in Section 3.6.2. Basically, our experiment with $\tau_{exp} = 2.5$h simulates a low precipitation rate situation resulting in an estimated mean effective transport duration of $\tau_t \approx 5$ min (Section 3.6.2) for individual snow particles, while the experiments with $\tau_{exp} = 0.5$h simulate a high precipitation rate situation with $\tau_t \approx 1$ min.*

*However, we agree that the current separation into precipitation and no-precipitation events results in a grey zone where processes may overlap, and that new temporally highly resolved models should aim for more physically based descriptions of these processes. Therefore, a particle shape-based parameterization of the density and SSA as proposed in Crocus (Vionnet et al., 2012) based on dendricity and sphericity in combination with the effective transport duration $\tau_t$ would likely be the way to go, although these parameters are very difficult to quantify or verify experimentally. This is probably the main reason why current snow models prefer using simple empirical correlations instead of physically based process descriptions.*

*We will include the above discussion and your suggestions accordingly in the revised manuscript in Section 3.4.2. We also redefine the parameters in Eq. 1 and 2, where the snow density $\rho_s$ will be renamed as $\rho_{ns}$ (new snow density), and clearly state here and elsewhere that we only consider new snow densification under the influence of wind. We will rewrite these equations as*

$$\rho_{ns} = 10^{\beta_0 + \beta_1 T_a + \beta_2 \arcsin(\sqrt{RH}) + \beta_3 log_{10}(V)} \qquad\qquad Eq.\ 1$$

$$\rho_{ns} = 50 + 1.7(T_{wb} - 258.16) + D_1 + D_2 \left[1 - e^{-D_3(V-5)}\right] \qquad Eq.\ 2$$

One of my concerns with the current evaluation shown on Figure 8 is that the authors derive densification rates from parameterizations of snowfall density that does not include any temporal component. These parameterizations only provide a value of snowfall density under given meteorological conditions, but they do not specify which time is needed to reach this value (especially for the wind contribution). It would be very valuable to go back to the original datasets that were used to derive these parameterizations and to better understand the temporal aspect. For example, if these values were derived from measurements of snow taken on snow board, are they representative of 1, 3, 6 or 12, 24 hours snow accumulation? Maybe, it could help to understand the large differences in snow density resulting from these parameterizations. In addition, in Fig. 8, which value is taken when computing the initial ice volume fraction for the parameterizations?

*Generally, as discussed in Section 3.6.2, the temporal component is extremely difficult to quantify experimentally, because the important timescale is the mean effective transport duration $\tau_t$ of the snow particles which defines the magnitude of fragmentation and airborne metamorphism (see Section 3.6.2). This time $\tau_t$ depends mainly on the particle size and shape, wind speed and the precipitation intensity, where the latter defines how fast additional precipitation particles will cover up the underlying deposited snow which then cannot be re-mobilized again.*

*The parameterization of the SNOWPACK model (Lehning et al. 2002) is based on field measurements at the Weisfluhjoch (WFJ) field site in Davos, Switzerland. The density of the topmost 3 cm of new snow were regularly sampled during different snowfall events, resulting in a maximum temporal resolution of one sample per 0.5 h depending on the precipitation intensity. This is comparable to the timescales of our experiments with $\tau_{exp} = 0.5 - 2.5$ h. We did not find any information on the timescales of the dataset used in SnowTran-3D (Liston et al., 2002) for the parameterization of Eq. 2 above, i.e. the parameters $D_1$-$D_3$. However, we assume a daily measured new snow density has been correlated to a 24h mean wind speed for days with precipitation, where intermittent snowfall and variable wind speeds may strongly affect the results. A significant contribution of densification without precipitation during these days may thus result in longer effective transport durations $\tau_t$ and thus higher densification rates as shown in Fig. 8 for the SnowTran-3D model, that could potentially explain the large model discrepancies. We will include this discussion in the revised version in Section 2.4.*

*Regarding the initial ice volume fraction, we wrote in L215: "The initial new snow densities representing snow deposited without wind ($V_{0.4m} = 0$ m s$-1$) measured inside the Snowmaker box were ranging in between 45-80 kg m$-3$ (Fig. 6a-b) ..." we will add the information that these densities define the initial ice-volume fraction $\Phi_{i0}$ for the calculation of the densification rates in the following figures.*

Overall, there is no doubt that the data collected in this study will inform the development of improved compaction routines due to aeolian snow transport.

*Thank you very much!*

**Specific Comments**

P1 L9: Add at which height were taken the wind speed measurements?

*The wind speed was measured at a height of 0.4 m. We will add this information accordingly in the abstract.*

P1 L10-12: as mentioned above, the main contribution of this paper is to provide a set of very rich measurements to better understand the impact of the wind on the properties of surface snow. I recommend the authors to highlight first in the abstract the main conclusions derived from these measurements before mentioning the comparison with existing parameterizations.

*We agree and will rearrange the abstract accordingly.*

P1 L 18-22: at this stage, the introduction lacks clarity. I recommend the authors to make a better distinction between blowing snow events with and without concurrent snowfall and to describe the types of particles that are transported in these two situations.

*We agree. Therefore, we will make a clear distinction as suggested and add more information on the particle types as discussed above (dendritic new snow during precipitation and typically more rounded grains in the absence of precipitation).*

P2 L 45-46: the author can refer here to Royer et al. (2021), Wever et al (2022) and Amory et al (2021) to illustrate how the parameterizations of the increase of surface snow density due to wind can be adjusted to better represent the properties of surface snow in the Arctic and in Antarctica.

*We agree that this is a good idea, and we will add these references.*

P3 L 85-90: Section 2.2 explains the detail of the different experiments. Before jumping straight into the detailed description of the experiments, it would be good for the reader to give an overview of what is tested with these 12 experiments.

*We agree that we should first mention that we want to test different wind speeds, temperatures, and transport durations with these 12 experiments and that the experiment #13 aims at measuring particle impact characteristics with the high-speed camera and experiment #14 is a sensitivity experiment to estimate the magnitude of airborne snow metamorphism.*

P 5 P 111-112: at which heights are measured the air temperature and relative humidity in the RWT.

*The air temperature and relative humidity are measured at a height of 15 cm. We will add this information in Section 2.3.*

P 6 L 137: Eq. 1 is not described in Lehning et al. (2002). It seems that the ZWART equation has been developed later. Can the authors add a reference? The equation is also described in this supplementary material (Section 4 of https://tc.copernicus.org/articles/17/519/2023/tc-17-519-2023-supplement.pdf).

*Great! Thanks! We used the Lehning et al. (2002) reference as a general reference for the SNOWPACK model, as the ZWART parameterization was never published officially. We will use this very recent reference now.*

P 6 L 146-148: If the authors manage to correctly justify why they are evaluating parameterization of falling snow density, the parameterization of Pahaut (1975) implemented in Crocus (Vionnet et al., 2012) could be tested as well. Indeed, it only depends on temperature and wind speed.

*Thanks a lot for this important comment! We hope we convinced you in the above discussion on transport durations and snow types that we should only consider parameterizations with concurrent snowfalls. We somehow missed the parameterization introduced in Vionnet et al. (2012) (Eq. 1) and tested it now as well:*

$$\rho_{ns} = a_\rho + b_\rho(T_a - T_{fus}) + c_\rho\sqrt{V} \qquad\qquad Eq.\ 3$$

[Figure]

*We obtain the following parameters $a_\rho = 43$ kg m$^{-3}$, $b_\rho = 9$ kg m$^{-3}$ K$^{-1}$ and $c_\rho = 35$ kg m$^{-7/2}$ s$^{-1/2}$ when fitting Eq. 3 to our data. Minimum air temperatures of $T_a = -10°C$ (Crocus) and $T_a = -15°C$ (SnowTran-3D) had to be used for these model fits instead of the actual air temperature $T_a = -24°C$ measured during the Experiments 1-7 (Table 1), because the models do not result in realistic values for low air temperatures. However, Fig. 9b in our manuscript shows that the densification rate tends to be temperature independent below approximately $T_a < -6°$. Therefore, the fit parameters for the crocus model are only considered to be valid for $V_{0.4m} > 3.8$ m s$^{-1}$ in Eq. 3, which corresponds to a wind speed of 5 m s$^{-1}$ at a height of 2m with an aerodynamic roughness length of $z_0 = 0.24$ mm for fresh snow as determined by Gromke et al. (2011). We will introduce the Crocus parameterization in Section 2.4 as well as the results and discussion in Section 3.4.2.*

P 7 L 168-170: In this paragraph, the authors measure if the particle impact characteristics in the RWT are consistent with natural conditions. They compare their results with the measurements from Sugiura et al . (2000). However, these measurements were also collected in a wind tunnel. Can the authors elaborate on the definition of natural conditions?

*We agree that "natural conditions" is misleading. What we mean is "well developed boundary layer flow conditions" as it can be achieved in a linear wind tunnel as used by Sugiura et al. (2000). We will change this accordingly.*

P 10 L 198-200: could the authors test the statistical significance of the regression lines shown on Fig. 5a, 5b and 5c?

*Great question! The p-values for the statistical significance are $p = 0.011$ (strong evidence, Fig. 5a), $p = 0.079$ (weak evidence or trend, Fig. 5b), and $p = 0.0067$ (strong evidence, Fig. 5c). We will add this information in Section 3.2.*

P 15 L 305-308: it would make sense to propose a fit that respects the physical grounds and tends to zero for very long times.

*We did this previously. However, because of the limited data available (only 3 measurement points, Fig. 9a), a simple two-parameter function is required. A two-parameter exponential function results in a bad representation (almost linear in this range) of the data. Based on the definition of the densification rate, a reciprocal function seemed to be most feasible resulting in a reasonable fit to the data in the measurement range. Therefore, we would rather keep it as simple as it is with the statement we provided: "… simply to represent the data points in the experimental range. On physical grounds, the densification rate should tend to zero for very long times." We would also add: "The good fit of the reciprocal function indicates that the time (experiment duration) governs the decrease of the densification rate and not the change in ice volume fraction."*

P 18 L 370-371: it would be interesting to mention that the effect of ambient relative humidity should be tested as well due to its large impact on blowing snow sublimation.

*This is a good point and we will mention this in the revised version.*

P 20 L 419: would it be possible to write Eq (7) in terms of SSA rate as the previous equations?

*We thought about this. However, the different time scale used in this additional sensitivity experiment (the mean effective "transport duration" $\tau_t$) we want to avoid the definition of an additional SSA rate that can easily be confused with the ones from Fig. 10 and 11 which are based on the experiment duration $\tau_{exp}$.*

**Technical Comments**

**Figures**

Figure 3: can the authors add on the three photos the corresponding time stamps as well as a vertical and horizontal scale?

[Figure]

*Good point! We added time stamps and a scale.*

Figure 6: it would be interesting to have the same range of values for the y-axis of Fig 6c and 6d. Otherwise, it seems stronger SSA decreased are measured with the mico-CT.

*We had this initially, however, the micro-CT data becomes too compressed so that the individual measurements cannot be clearly identified anymore. We would rather keep it as it is and put a note there that the reader should be aware of the different y-axis scales.*

Figure 6: If one micro_CT measurement has been collected for each experiment, what does represent the error bars shown on Fig 6b and d?

*Thanks for this important comment! The error bars for the micro-CT measurements in Fig. 6b,d are one standard deviation of the vertical density and SSA variability of the 3-6 cm high snow samples with 20-40 data points. We will add this information in Section 3.3*

**Tables**

Table 1: mention in the caption if relative humidity is measured with respect to ice.

*Good point! We will include this in the caption!*

Table 1: it would be interesting to know on this table for which experiments micro-CT measurements have been carried out.

*Good point! We added a column in the table below:*

**Table 1.** Overview of the experimental settings and atmospheric conditions for the main experiments (1-12) and the complementary experiments (13-14). The average value for $RH$ with respect to ice is calculated from the second period of each experiment where a situation close to equilibrium for $RH$ is reached (Fig. 2).

| Experiment | Mean wind speed $V_{0.4m}$ [$ms^{-1}$] | Experiment duration $\tau_{exp}$ [h] | Average air temperature $T_a$ [°C] | Average relative humidity $RH$ [%] | $\mu CT$ measurements yes / no |
|---|---|---|---|---|---|
| 1 | 5.0 | 0.5 | -24.0 | 92.0 | no |
| 2 | 6.9 | 0.5 | -24.6 | 99.5 | no |
| 3 | 6.0 | 0.5 | -23.8 | 99.5 | no |
| 4 | 7.1 | 0.5 | -21.3 | 99.5 | no |
| 5 | 4.0 | 0.5 | -20.6 | 98.6 | yes |
| 6 | 6.6 | 0.5 | -20.6 | 98.7 | yes |
| 7 | 5.0 | 0.5 | -23.1 | 98.5 | yes |
| 8 | 6.0 | 1.0 | -21.7 | 98.1 | yes |
| 9 | 6.0 | 2.5 | -21.0 | 100.7 | yes |
| 10 | 6.0 | 0.5 | -11.5 | 100.5 | yes |
| 11 | 6.0 | 0.5 | -5.6 | 99.9 | yes |
| 12 | 6.0 | 0.5 | -2.4 | 99.4 | yes |
| 13 | 3.0 - 7.0 | 5.8 | -20.6 | 83.5 | no |
| 14 | 7.9 | 2.5 | -18.0 | 98.5 | yes |

**References (used in this review and not present in the initial manuscript)**

Amory, C., Kittel, C., Le Toumelin, L., Agosta, C., Delhasse, A., Favier, V., & Fettweis, X. (2021). Performance of MAR (v3. 11) in simulating the drifting-snow climate and surface mass balance of Adélie Land, East Antarctica. *Geoscientific Model Development*, *14*(6), 3487-3510.

Pahaut, E.: La métamorphose des cristaux de neige (Snow crystal metamorphosis), Monographies de la Météorologie Nationale, Vol. 96, Météo France, 1975.

Royer, A., Picard, G., Vargel, C., Langlois, A., Gouttevin, I., & Dumont, M. (2021). Improved simulation of arctic circumpolar land area snow properties and soil temperatures. Frontiers in Earth Science, 9, 685140.

Wever, N., Keenan, E., Amory, C., Lehning, M., Sigmund, A., Huwald, H., & Lenaerts, J. T. (2023). Observations and simulations of new snow density in the drifting snow-dominated environment of Antarctica. *Journal of Glaciology*, *69*(276), 823-840.

*Thanks! We add these references at the specific locations.*

---

## Author Response (AR1)

Dear Editor,

please find below the changes that were made based on the reviewer comments. Small additional changes were made to the manuscript for the benefit of consistency and readability. These changes are not included in this document but are highlighted in the marked-up manuscript version.

With best regards,
Benjamin Walter

**RC1**: 'Comment on tc-2023-112', Anonymous Referee #1, 15 Sep 2023

1) The authors present a very exciting and compelling experiment focused on the effect of wind snow surface microstructure. For how simple the question is, this is an incredibly hard problem to work on. We have limited tools at our disposal to make concise measurements of snow microstructure, and it is incredibly difficult to run such an experiment in the field. This group at SLF has succeeded at combining their expertise in snow microstructure and wind tunnel experiments to provide new insights into this intriguing aspect of snow metamorphism.

2) My only concern with this research is the significantly unphysical conditions under which snow is being transported. As it stands, I do not see a reason why the rate of change of any snow characteristics in their experiment should be related to any measurements of natural snow undergoing natural transport.

We added the following sentences in Section 5, L511:

*"The focus of this study is on linking atmospheric and aeolian snow transport conditions during precipitation events to the snow microstructure of the ultimately deposited snow and to identify the relevant processes. While we show magnitudes of dependencies between different flow and snow parameters, processes and snow microstructures, natural conditions may be different in the field, depending on the snow type or flow conditions, while the latter is also rarely well developed and stationary for natural conditions."*

3) It is unclear if the snow particles actually come in contact with the propeller driving their RWT. A schematic that shows this mechanism would be very helpful.

We added the following sentences in Section 2.2, L102:

*"The wind turbine propeller is located beneath the lid in the curved section of the ring wind tunnel (Fig. 1a) covering approximately the top quarter of the wind tunnel cross-section. The propeller blades are 90 mm long. As the particle mass flux exponentially decreases with height as shown by Yu et al. (2023) for our RWT, only a negligible part of the snow particles will get into contact with the propeller. A schematic drawing of the RWT and additional figures can be found in Yu et al. (2023)."*

4) More importantly, the authors acknowledge that a large portion of snow particles are transported along the outer wall due to centrifugal forces and, among other effects, this causes a measurable impact on density. This is well outside the realm of normal saltation and suspension. Given that $v_x$ is so much larger than $v_z$, this impact force may be considerably higher than in nature. As well, repeat impacts caused by snow working its way around a corner may cause orders of magnitude more fragmentation.

*We added the following sentences at the end of Section 3.4.1, L317:*

*"In the curved sections, particles are mainly sliding along the RWT outer wall, an effect that is certainly not favourable for simulating natural snow transport but inevitable for a compact closed circuit wind tunnel in a cold laboratory of limited dimensions. The centrifugal forces acting on snow particles in the curved section were estimated being two to three orders of magnitude smaller compared to the forces acting on the snow particles during surface impact while saltating. The maximum impact angles of snow particles impacting into the curved outer wall were calculated being around 25°-30° which are comparable to the impact angles $\alpha_{in}$ on the snow surface in the straight test section (Fig. 4b and d). However, a Stokes number < 0.1 indicates a good flow following behaviour of the snow particles when the air flow gets redirected in the curved section, resulting in smaller impact angles. Based on these results, we conclude that both, centrifugal forces, and particle impacting into the curved side walls have at most a similar but more likely a smaller effect on particle fragmentation than particles impacting on the surface during saltation. Based on these estimates, it can be assumed that particle fragmentation is dominated by the particle impacts on the snow surface in the saltation layer, and that our results are, to a certain degree, comparable to real natural snow transport situations."*

Given these concerns, could you please address the question of transport around the curves (impact velocities, momentum balance, fragmentation rate, restitution coeff, how many more impacts per second? etc.), or modify the manuscript in such a way that the reader knows while you may have novel measurements of a physical process, this physical process has little relation to what one may expect to find in nature? As it stands, I think the quantitative information provided needs to be qualified or better justified.

There are a few grammatical things that could be improved:

L8- Cover wind speeds? *We changed this to: "vary wind speeds".*

L11-In the deposit? *We changed this to: "deposited snow".*

L21- Is rolling different from creep? *No: we call it now "rolling or creeping"*

L30-Chemical species? *We changed this to "Chemical substances".*

L64: Do you mean necessary or inevitable? *We changed this to "necessary".*

L194: To make contact to previous studies? *We changed this to: "To compare the quality of our particle transport phase to previous studies".*

Other comments

L56: At what height are these wind velocities? *We added the information that the wind speed was measured at a height of h = 1 m.*

L76-77: Very cool Thanks.

L80-82: Do the particles not come in contact with the propeller?

*We added additional information in Section 2.2 (please see your comment 3 above).*

Figure 2: How did you conclude the jump in RH was from snow particle sublimation? What's the RH of the cold room?

*We added the following sentences at the end of Section 3.1, L183:*

*"The relative humidity of the cold room varies depending on how often people enter and leave the room during a day. In the morning, the RH is typically low at around 40% - 50%. Therefore, depending on the initial RH before an experiment, a more or less strong increase in RH is obtained due to sublimation of the suspended particles (Dai and Huang, 2014)."*

L258-259: Very cool *Thanks.*

L267-268: Again, how can you decouple this from the effect of particles smashing into walls that are necessarily there in nature?

*We added additional information on this at the end of Section 3.4.1 (please see your comment 4 above).*

**RC2**: 'Comment on tc-2023-112', Anonymous Referee #2, 15 Sep 2023

**Review of the paper "Wind tunnel experiments to quantify the effect of aeolian snow transport on the surface snow microstructure" by Walter et al. submitted to The Cryosphere.**

This paper presents an innovative set of measurements to investigate the effect of wind-induced snow transport on the physical properties of surface snow (density, SSA). These measurements were collected in a ring-shaped wind tunnel (RWT) that reproduces the main characteristics of aeolian snow transport. The authors quantified the changes in density and SSA during events with different wind speed and air temperature. Their analysis confirmed the increase in surface snow density with increasing wind speed that has been observed in the field and highlighted a slight decrease in SSA with increasing wind speed. The author also compared the densification rates measured in the RWT with parameterizations used in snowpack schemes.

The subject of this paper is very interesting for the snow community and presents a set of original measurements to quantify the effects of wind on the physical properties of surface snow. So far, these quantifications have mainly been obtained from field measurements (mainly for surface snow density) that are influenced by other physical processes, making it challenging to disentangle the effect of the wind from the other processes. These measurements can serve to develop more-physically based parameterizations of the impact of wind on the physical properties of the snow cover in multi-layer snowpack schemes such as Crocus and SNOWPACK. Therefore, this paper should be published in The Cryosphere. However, prior to publication, the author must carefully define in which context they are working (blowing snow event with concurrent snowfall) and revise accordingly which existing parameterizations they are evaluating in the context of the study. These two general comments are followed by more specific and technical comments.

**General comments**

1. In this paper, the authors study the effects of aeolian snow transport on the properties of surface snow using a RWT where snow is continuously added to mimic snow precipitation until the end of the different experiments. In this respect, the authors are reproducing in their experiments what happened during blowing snow events with concurrent snowfall. In such conditions, most of the snow transported by the wind is made of precipitating snow particles that fall continuously on the snow surface and are then transported by the wind as illustrated nicely by Figure 3 in the submitted paper. Snow particles initially present at the snow surface may also be transported if the wind speed is sufficient. This situation differs from blowing snow events without snowfall when the transported snow is only made of snow initially present at the snow surface without the constant supply of new snow particles from snowfall. Even if the physical processes involved are the same in terms of particle impacts and ejections, we can expect different densification rates and changes in SSA due the influence of the constant supply of fresh dendritic snow during blowing snow events with concurrent snowfall. Therefore, I strongly recommend to the authors to define well the type of blowing events that they are simulating in the wind tunnel and to discuss how the constant supply of fresh snow influences the results presented in their study. Without such clear discussion, their conclusions may be applied erroneously to different situations that were not captured yet in the RWT experiments.

We agree that we need to better highlight (especially in the abstract) that we only consider blowing snow events with concurrent snowfall. We revised the manuscript accordingly:

First sentence of the abstract L1: *"The evolution of the surface snow microstructure under the influence of wind during precipitation events is hardly understood but crucial for polar and alpine snowpacks."*

Middle of the abstract L8: *"We used a single snow type (dendritic fresh snow) for simulating precipitation, …"*

Section 1, L67: *"It is the aim of our study to propose an experimental setup to systematically investigate how wind affects the evolution of the surface snow density and SSA during precipitation events as functions of the wind speed, air temperature and transport duration."*

Already existing information in Subsection 2.2 L96: *"… , temporally equally distributed over the entire experiment duration $\tau_{exp}$ (Table 1), to mimic snow precipitation until the end of the experiment."*

2. The authors have evaluated two parameterizations used in snowpack schemes for snow densification that account for the effects of wind. This part of the paper is very valuable for snowpack schemes, but it should be strongly revised to reflect well how the effects of wind on surface snow properties are included in snowpack schemes and then to make sure that the correct parameterizations are tested in this study.

The effect of wind in snowpack schemes such as Crocus, SNOWPACK and SNOWMODEL consist in a two-step process: (i) the falling snow density generally includes a dependency on wind speed (Pahaut, 1975; Lehning et al., 2002; Liston et al., 2007). It is computed at each model time step based on the meteorological forcing (wind speed, air temperature, …) and new snow is added at the top of the snowpack, (ii) the models then account for wind-driven compaction by including a wind compaction term when calculating compaction in the near surface snow layers (Brun et al., 1997; Liston et al., 2007; Amory et al., 2021; Wever et al., 2022). The wind compaction rate depends on the intensity of aeolian snow transport. This second component allows the models to simulate the increase in surface density during blowing snow events without snowfall (cf my first general comment). So far, in their paper, the authors are evaluating two parameterizations for the density of new snow and are ignoring the second component of wind-driven compaction in numerical snow models. I recommend them to better justify why they are only evaluating the first component. There is a clear grey zone in between these two model components with parameterizations that may overlap and may treat twice the same physical process. Ultimately, we could imagine a clearer separation in snowpack schemes where (i) the snowfall density only depends on temperature and density (to indirectly represent the variability in falling hydrometeors) and (ii) all the wind-related process are all treated by a wind-compaction routine. Under this assumption, the measurements presented in this paper would serve to adjust such wind-compaction routines.

We added the following information in Section 2.4, L155: *"In all three models, snow densification by wind is separated in a new snow density term for describing snow compaction during precipitation events, and in a term describing wind-driven compaction during blowing snow events without precipitation. This separation is likely resulting from different time scales and snow types involved in these different processes. During precipitation events, the typically highly dendritic new snow will quite quickly (depending also on the wind speed) cover the underlying (new) snow which then can't be entrained anymore by the wind, resulting in rather short effective transport durations. However, without precipitation, loose surface snow of likely different grain types (potentially more decomposed rounded particles) may be entrained and affect by wind for much longer time (or transport) durations. As we simulate precipitation events in our RWT, only the equations for the new snow density with concurrent snowfall are considered in our study."*

We added the following information in Section 3.4.2, L356: *"The current separation into precipitation and no-precipitation events in the three models as discussed in Section 2.4 results in a grey zone where processes may overlap. New, temporally highly resolved models may aim for more physically based descriptions of these processes. Therefore, a particle shape-based parameterization of the density and SSA as proposed in CROCUS (Vionnet et al., 2012) based on dendricity and sphericity in combination with an effective particle transport duration $\tau_t$ would likely be beneficial to simultaneously cover precipitation and no-precipitation events in future modelling attempts. However, these parameters are very difficult to measure and quantify experimentally which is probably the reason why current snow models prefer using simple empirical correlations instead of physically based process descriptions."*

We redefined the parameters in Eq. 1 and 2, where the snow density $\rho_s$ is now renamed as $\rho_{ns}$ (new snow density):

$$\rho_{ns} = 10^{\beta_0 + \beta_1 T_a + \beta_2 \arcsin(\sqrt{RH}) + \beta_3 \log_{10}(V)} \qquad \text{Eq. 1}$$

$$\rho_{ns} = 50 + 1.7(T_{wb} - 258.16) + D_1 + D_2 \left[1 - e^{-D_3(V-5)}\right] \qquad \text{Eq. 2}$$

3. One of my concerns with the current evaluation shown on Figure 8 is that the authors derive densification rates from parameterizations of snowfall density that does not include any temporal component. These parameterizations only provide a value of snowfall density under given meteorological conditions, but they do not specify which time is needed to reach this value (especially for the wind contribution). It would be very valuable to go back to the original datasets that were used to derive these parameterizations and to better understand the temporal aspect. For example, if these values were derived from measurements of snow taken on snow board, are they representative of 1, 3, 6 or 12, 24 hours snow accumulation? Maybe, it could help to understand the large differences in snow density resulting from these parameterizations. In addition, in Fig. 8, which value is taken when computing the initial ice volume fraction for the parameterizations?

We added the following information in Section 3.4.2, L335: *"The density of the topmost 3 cm of new snow were regularly sampled during different snowfall events, resulting in a temporal resolution of one sample per 0.5-1 h depending on the precipitation intensity. This is comparable to the timescale of our experiments with $\tau_{exp} = 0.5$ h (Fig. 8)."*

We added the following information in Section 3.4.2, L338: *"We did not find any information on the timescales and sampling of the data set used for the SnowTran-3D parameterization (Liston et al., 2002). However, the higher densification rates predicted by the SnowTran-3D model (Fig. 8) may result from different time scales, atmospheric conditions, or additional transport in the absence of precipitation involved in the measurements they used for their parameterization. The parameters in Eq. (3) for the CROCUS model originate from a study carried out by Pahaut (1976) at Col de Porte (1325m altitude, French Alps). Unfortunately, no information on these measurements could be found."*

We added the following information in Section 3.3, L248: *"The initial new snow densities representing snow deposited without wind ($V_{0.4m} = 0$ m s$-1$) measured inside the Snowmaker box were ranging in between 45-80 kg m$-3$ (Fig. 6a-b) and define the initial ice-volume fraction $\Phi_{i0}$ for the calculation of the densification rates in the following figures."*

Overall, there is no doubt that the data collected in this study will inform the development of improved compaction routines due to aeolian snow transport.

**Specific Comments**

P1 L9: Add at which height were taken the wind speed measurements?

We added the following information in the abstract, L8: "…vary wind speeds at a height of 0.4 m from 3 m s-1 to 7 m s-1 …"

P1 L10-12: as mentioned above, the main contribution of this paper is to provide a set of very rich measurements to better understand the impact of the wind on the properties of surface snow. I recommend the authors to highlight first in the abstract the main conclusions derived from these measurements before mentioning the comparison with existing parameterizations.

We rearranged the abstract accordingly, L10: *"The measured airborne impact trajectories confirm the consistency of our coefficient of restitution with large scale saltation, rendering the setup suitable to realistically study interactions between airborne and deposited snow. Increasing wind speeds were found to result in intensified densification and stronger SSA decreases. The most drastic snow density and SSA changes of deposited snow are observed close to the melting point. Our measured densification rates as a function of wind speed show clear deviations from existing statistical models but can be re-parameterized through our data."*

P1 L 18-22: at this stage, the introduction lacks clarity. I recommend the authors to make a better distinction between blowing snow events with and without concurrent snowfall and to describe the types of particles that are transported in these two situations.

Please see the answer to your comment 3 above.

P2 L 45-46: the author can refer here to Royer et al. (2021), Wever et al (2022) and Amory et al (2021) to illustrate how the parameterizations of the increase of surface snow density due to wind can be adjusted to better represent the properties of surface snow in the Arctic and in Antarctica.

*We added this information in the Introduction Section 1, L47: "However, Royer et al. (2021), Amory et al. (2021), and Wever et al. (2022) illustrate how the parameterizations of the increase of surface snow density due to wind can be adjusted in these models to better represent the properties of surface snow in the Arctic and in Antarctica."*

P3 L 85-90: Section 2.2 explains the detail of the different experiments. Before jumping straight into the detailed description of the experiments, it would be good for the reader to give an overview of what is tested with these 12 experiments.

*We added the following information in Section 2.2, L89: "We performed a total of 14 RWT experiments for testing the effects of wind speed (experiment 1-7), transport duration (experiments 8-9) and temperature (experiments 10-12) on the surface snow microstructure (Table 1). Particle impact characteristics were measured with a high-speed camera during experiment 13, while experiment 14 served as a sensitivity study to test the effect of sublimation and vapor re-deposition on airborne particles.*

P 5 P 111-112: at which heights are measured the air temperature and relative humidity in the RWT.

*We added the following information in Section 2.3, L125: "The air temperature and relative humidity are measured at a height of 0.15 m, while the wind speed is measured at a height of 0.4 m."*

P 6 L 137: Eq. 1 is not described in Lehning et al. (2002). It seems that the ZWART equation has been developed later. Can the authors add a reference? The equation is also described in this supplementary material (Section 4 of https://tc.copernicus.org/articles/17/519/2023/tc-17-519-2023-supplement.pdf).

*We added the following reference in Section 2.4, L163: "Zwart, C. (2007) Significance of new-snow properties for snow-cover development, Master thesis, WSL Institute for Snow and Avalanche Research SLF, Davos, Switzerland."*

P 6 L 146-148: If the authors manage to correctly justify why they are evaluating parameterization of falling snow density, the parameterization of Pahaut (1975) implemented in Crocus (Vionnet et al., 2012) could be tested as well. Indeed, it only depends on temperature and wind speed.

*We introduced the parameterization from Vionnet et al. (2012) (Eq. 1) in Section 2.4 and tested it now as well in Section 3.4.1 and 3.4.2:*

*Section 2.4, L173: "Another empirical description for the new snow density affected by wind during precipitation events is implemented in the CROCUS snow model (Brun et al., 1997; Vionnet et al., 2012) which is also expressed as a function of the wind speed V and the air temperature $T_a$:*

$$\rho_{ns} = a_\rho + b_\rho(T_a - T_{fus}) + c_\rho\sqrt{V} \qquad\qquad Eq.\ 3$$

*In Eq. 3, $T_{fus}$ is the temperature of the melting point for water, $a_\rho = 109\ kg\ m^{-3}$, $b_\rho = 6\ kg\ m^{-3}\ K^{-1}$ and $c_\rho = 26\ kg\ m^{-7/2}\ s^{-1/2}\ s{-}1/2$ are constants."*

New Fig. 8 including the CROCUS model results:

[Figure]

Section 3.4.1, L301: *"Furthermore, Fig. 8 includes model predictions and model fits to our RWT data using the theoretical descriptions from the SNOWPACK, SnowTran-3D and CROCUS models (Eq. 1-3)."*

Section 3.4.2, L348: *"The following parameters $a_\rho = 43\ kg\ m^{-3}$, $b_\rho = 9\ kg\ m^{-3}\ K^{-1}$ and $c_\rho = 35\ kg\ m^{-7/2}\ s^{-1/2}$ were obtained for the CROCUS model when fitting Eq. 3 to our data. Minimum air temperatures of $T_a = -10°C$ (CROCUS) and $T_a = -15°C$ (SnowTran-3D) had to be used for these model fits instead of the actual air temperature $T_a = -24°C$ measured during the experiments 1-7 (Table 1), because the models do not result in realistic values for lower air temperatures. However, Fig. 9b in our manuscript shows that the densification rate tends to be temperature independent below approximately $T_a < -6°$. Therefore, the fit parameters for the CROCUS model are only considered to be valid for $V_{0.4m} > 3.8\ m\ s^{-1}$ in Eq. 3, which corresponds to a wind speed of 5 m s$^{-1}$ at a height of 2m with an aerodynamic roughness length of $z_0 = 0.24\ mm$ for fresh snow as determined by Gromke et al. (2011). "*

P 7 L 168-170: In this paragraph, the authors measure if the particle impact characteristics in the RWT are consistent with natural conditions. They compare their results with the measurements from Sugiura et al . (2000). However, these measurements were also collected in a wind tunnel. Can the authors elaborate on the definition of natural conditions?

We replaced "natural conditions" with "well-developed boundary layer flow" in Section 3.2, L200: *"We thus argue here that the boundary layer flow may not necessarily be perfectly homogeneous, stationary, and well-developed, as long as the particle impact characteristics are consistent with those of a well-developed boundary layer flow as studied by Sugiura et al. (2000)"*

P 10 L 198-200: could the authors test the statistical significance of the regression lines shown on Fig. 5a, 5b and 5c?

*We added the following information in Section 3.2, L233: "The p-values for the statistical significance are p = 0.011 (strong evidence, Fig. 5a), p = 0.079 (weak evidence or trend, Fig. 5b), and p = 0.0067 (strong evidence, Fig. 5c)."*

P 15 L 305-308: it would make sense to propose a fit that respects the physical grounds and tends to zero for very long times.

We tested such a fit previously. However, because of the limited data available (only 3 measurement points, Fig. 9a), a simple one or two-parameter function is required. Simply removing the parameter $B_1$ from Eq. 5 (previously Eq. 4) results in a poor fit. A two-parameter exponential function also results in a bad representation (almost linear in this range) of the data. Based on the definition of the densification rate, the reciprocal function seemed to be most feasible resulting in a reasonable fit to the data in the measurement range. Therefore, we would rather keep it as simple as it is and added and modified the following sentences in Section 3.4.3, L380: *"... with $A_1 = 1.30$ and $B_1 = 0.80 \ h^{-1}$, simply to represent the three data points in the experimental range with a two-parameter fit. On physical grounds, the densification rate should tend to zero instead for very long times, which would require more data points to obtain a reasonable fit. However, the good fit of the reciprocal function (Eq. 5) indicates that the time $\tau_{exp}$ (experiment duration) governs the decrease of the densification rate and not the change in ice volume fraction."*

P 18 L 370-371: it would be interesting to mention that the effect of ambient relative humidity should be tested as well due to its large impact on blowing snow sublimation.

We added this information in Section 3.5.2, L447: *"Whether the proposed parameterization of Eq. 6 is valid for different wind speeds $V_{0.4m}$, experiment durations $\tau_{exp}$ and relative humidity RH must be tested in future studies."*

P 20 L 419: would it be possible to write Eq (7 -> now 8) in terms of SSA rate as the previous equations?

We thought about this. However, the different time scale used in this additional sensitivity experiment, the mean effective "transport duration" $\tau_t$ instead of the experiment duration $\tau_{exp}$, we want to avoid the definition of an additional SSA rate that can easily be confused with the ones from Fig. 10 and 11 which are based on $\tau_{exp}$.

**Technical Comments**

**Figures**

Figure 3: can the authors add on the three photos the corresponding time stamps as well as a vertical and horizontal scale?

We added time stamps and a millimetre scale to Fig. 3:

[Figure]

Figure 6: it would be interesting to have the same range of values for the y-axis of Fig 6c and 6d. Otherwise, it seems stronger SSA decreased are measured with the mico-CT.

*We had this initially, however, the micro-CT data (Fig. 6d) becomes too compressed, so that the individual measurements and the SSA reduction cannot be clearly identified anymore. We keep it as it is and put a note in the description of Fig. 6 that the reader should be aware of the different y-axis scales:* "*A different y-scaling is used in d) relative to c) for a better visualization of the SSA changes for the μCT measurements.*"

Figure 6: If one micro_CT measurement has been collected for each experiment, what does represent the error bars shown on Fig 6b and d?

*We added the following sentence in Section 3.3, L267:* "*The error bars for the micro-CT measurements in Fig. 6b and 6d are derived from the standard deviation of the vertical density and SSA variability of the 3-6 cm high snow samples with 20-40 data points.*

**Tables**

Table 1: mention in the caption if relative humidity is measured with respect to ice.

*We added this information in the caption of Table 1:* "*The average value for RH measured with respect to ice is calculated from the second period of each experiment where a situation close to equilibrium for RH is reached (Fig. 2).*"

Table 1: it would be interesting to know on this table for which experiments micro-CT measurements have been carried out.

We added this information in Table 1:

**Table 1.** Overview of the experimental settings and atmospheric conditions for the main experiments (1-12) and the complementary experiments (13-14). The average value for $RH$ measured with respect to ice is calculated from the second period of each experiment where a situation close to equilibrium for $RH$ is reached (Fig. 2).

| Experiment | Mean wind speed $V_{0.4m}$ [$ms^{-1}$] | Experiment duration $\tau_{exp}$ [$h$] | Average air temperature $T_a$ [°C] | Average relative humidity $RH$ [%] | $\mu CT$ measurements yes / no |
|---|---|---|---|---|---|
| 1 | 5.0 | 0.5 | -24.0 | 92.0 | no |
| 2 | 6.9 | 0.5 | -24.6 | 99.5 | no |
| 3 | 6.0 | 0.5 | -23.8 | 99.5 | no |
| 4 | 7.1 | 0.5 | -21.3 | 99.5 | no |
| 5 | 4.0 | 0.5 | -20.6 | 98.6 | yes |
| 6 | 6.6 | 0.5 | -20.6 | 98.7 | yes |
| 7 | 5.0 | 0.5 | -23.1 | 98.5 | yes |
| 8 | 6.0 | 1.0 | -21.7 | 98.1 | yes |
| 9 | 6.0 | 2.5 | -21.0 | 100.7 | yes |
| 10 | 6.0 | 0.5 | -11.5 | 100.5 | yes |
| 11 | 6.0 | 0.5 | -5.6 | 99.9 | yes |
| 12 | 6.0 | 0.5 | -2.4 | 99.4 | yes |
| 13 | 3.0 - 7.0 | 5.8 | -20.6 | 83.5 | no |
| 14 | 7.9 | 2.5 | -18.0 | 98.5 | yes |

**References (used in this review and not present in the initial manuscript)**

We added these references to the bibliography:

Amory, C., Kittel, C., Le Toumelin, L., Agosta, C., Delhasse, A., Favier, V., & Fettweis, X. (2021). Performance of MAR (v3. 11) in simulating the drifting-snow climate and surface mass balance of Adélie Land, East Antarctica. *Geoscientific Model Development*, *14*(6), 3487-3510.

Pahaut, E.: La métamorphose des cristaux de neige (Snow crystal metamorphosis), Monographies de la Météorologie Nationale, Vol. 96, Météo France, 1975.

Royer, A., Picard, G., Vargel, C., Langlois, A., Gouttevin, I., & Dumont, M. (2021). Improved simulation of arctic circumpolar land area snow properties and soil temperatures. Frontiers in Earth Science, 9, 685140.

Wever, N., Keenan, E., Amory, C., Lehning, M., Sigmund, A., Huwald, H., & Lenaerts, J. T. (2023). Observations and simulations of new snow density in the drifting snow-dominated environment of Antarctica. *Journal of Glaciology*, *69*(276), 823-840.

---

## Author Response (AR2)

**Report #1**
**Submitted on 09 Feb 2024**
**Anonymous referee #2**

Summary and recommendation:

In the revised paper, the authors have addressed most of the questions that I raised previously about the type of blowing snow events considered in this study and the different model parameterizations that are tested against the wind tunnel data. I made below a suite of comments for the authors to consider to improve the description and evaluation of the model parameterizations. This paper is a very valuable contribution to the literature, and it should be published in TC.

Thanks a lot for your very valuable comments and for reviewing our article again.

Line comments (with line numbers referring to the new version of the paper, without Track Change mode):

P2 L 37: the range of density for strongly wind affected surface snow should be revised. Maybe the range 250 – 400 kg/m3 is more appropriate for polar snow. See for example:
- Fausto, R. S., Box, J. E., Vandecrux, B., Van As, D., Steffen, K., MacFerrin, M. J., ... & Braithwaite, R. J. (2018). A snow density dataset for improving surface boundary conditions in Greenland ice sheet firn modeling. Frontiers in Earth Science, 6, 51.
- Domine, F., Lackner, G., Sarrazin, D., Poirier, M., & Belke-Brea, M. (2021). Meteorological, snow and soil data (2013–2019) from a herb tundra permafrost site at Bylot Island, Canadian high Arctic, for driving and testing snow and land surface models. Earth System Science Data, 13(9), 4331-4348.

We adapted the density range and added the references. Fausto et al. (2018).

P 6 L 155-157: the description of the snow densification by wind in the models should be revised. Indeed, the wind-driven compaction represents the compaction of snow that has been previously deposited at the snow surface. It can be active with and without concurrent snowfall as soon as the wind condition are sufficient to generate ground-based snow transport. Therefore, the sentence "a term describing wind-driven compaction during blowing snow events without precipitation" is not accurate and should be revised.

Thanks, this information was indeed misleading, therefore we rephrased this sentence:
L155 "In all three models, snow densification by wind can be active with and without concurrent snowfall and is initiated when wind speed exceeds certain thresholds to generate snow transport. Therefore, all models include terms describing ground-based densification of surface snow layers due to wind transport, and new snow density terms ($\rho_{ns}$) that also depend on wind speed for describing an initial compaction of precipitation. "

P 7 L 175: Note that in Crocus the implementation of the Pahaut relationship assumes a minimal value of 50 kg/m3 for the snowfall density. This minimal value should certainly be taken into

account by the authors when deriving the optimal parameters for the Pahaut relationship (P 16 L345-355).

We added:
L179: "… and the minimal initial density is 50 kg m$^{2}$ (Pahaut, 1975)."

P 7 L 176: Royer et al. (2021) have proposed a modification of the Pahaut relationship to better represent the effect of wind on surface density in arctic environment. They double the value of c_rho. It could be interesting to test this alternative formulation in the paper and compare it with the fit proposed by the authors (P 16 L 345-355).

Thank you for this comment! We doubled the value c_rho according to Royer et al. (2021) as suggested which results in a significant increase of the absolute snow density but only in a small increase of the densification rate. The main discrepancy between the measurements and the models is, as you indicate in your next comment, the result of the estimated time scales involved.

P 15: on Figure 8 the authors derive densification rates from parameterizations of snowfall density that do not include any temporal component. These parameterizations only provide a value of snowfall density under given meteorological conditions. The author must explain how they have computed the densification rate from those parameterizations. This point was raised in my initial review and has not been properly addressed by the authors.

Thanks for asking again this important question. We seemed to have missed providing a direct answer to this question during the initial review! The answer was partially indirectly provided in Section 3.4.2, but your question exactly pinpoints the problem of involved time scales with recent parameterizations for the new snow density during precipitation events. Therefore, we added additional information in Section 3.4.2. New paragraph that also treats most of your comments below:

L366: "We conclude that the differences between the models and our measurements are mainly the result of the estimated time scale ($\Delta t$) used for the calculation of the densification rates (Fig. 8). The new snow densification parameterizations (Eq. 1-3) do not contain any temporal component at all, although the measurements they are based on involved some time scales. However, densification of new snow under wind during precipitation events not only depends on the wind speed, but also on an effective transport duration ($\tau_t$) of individual precipitation particles, which is mainly governed by the precipitation intensity and particle cohesion as discussed below. We used a time scale of $\Delta t$ = 0.5h for calculating the densification rates for our experiments and all three models (Fig. 8). This time scale is at least appropriate for the SNOWPACK model and our measurements. That the SNOWPACK model nevertheless predicts significantly lower densification rates might be the result of lower precipitation rates during their field measurements resulting in longer effective transport durations $\tau_t$ as discussed in the following Section (Fig. 9a). The discrepancy for the two other models (SnowTran-3D and CROCUS) is likely also the result of different time scales $\Delta t$ involved in their measurements used for the model parameterization. Changing $\Delta t$ from 0.5 h to 1 h for the SnowTran-3D model and to 0.1h for SNOWPACK and CROCUS already results in reasonable agreement of the models with our measurements, highlighting the strong dependency of the model on involved time scales. Additional

discrepancies between the model descriptions and our measurements may arise from the fact that we did not consider additional compaction of surface snow layers due to wind when using the models (Fig. 8), because our RWT simulations are similar to the field measurements used to parameterize the wind speed dependent new snow density in the models. This highlights the problem of overlapping processes, where wind compaction during precipitation may be treated twice in the models: Once within the description of the wind speed dependent new snow density (Eq. 1-3), and once during additional wind compaction of surface layers. We conclude that a clearer separation in snowpack schemes may improve future model attempts of wind induced snow compaction, where the snowfall density only depends on temperature and humidity (to indirectly represent the variability in falling hydrometeors) and all the wind-related processes are treated by a well calibrated wind-compaction routine. Overall, the discrepancies between the models and our measurements can be attributed to poorly defined time scales, different precipitation intensities, different initial precipitation particles, particle cohesion, and local topography and climate conditions. This highlights the importance for more detailed physical descriptions of snow densification."

P 16: L 356-357: As mentioned above, the models (at least Crocus) are not separating the wind densification into precipitation and no-precipitation events. During a blowing snow event with concurrent snowfall, both parameterizations (wind-dependent snowfall density and wind compaction routine) will contribute to the increase in surface density. It could explain why the formulations of Pahaut and Zwart used in Crocus and SNOWPCK cannot predict the observed densifications rates. They certainly need to be combined with a wind compaction routine to fully represent wind densification during blowing snow events with concurrent snowfall. It would be interesting to explicitly mention this feature in this part of the analysis.

We agree and included a discussion of this in the new paragraph (previous comment):
L378: "Additional discrepancies between the model descriptions and our measurements may arise from the fact that we did not consider additional compaction of surface snow layers due to wind when using the models (Fig. 8), because our RWT simulations are similar to the field measurements used to parameterize the wind speed dependent new snow density in the models. This highlights the problem of overlapping processes, where wind compaction during precipitation may be treated twice in the models: Once within the description of the wind speed dependent new snow density (Eq. 1-3), and once during additional wind compaction of surface layers."

But, as mentioned in my first review, this generates a clear grey zone in between these two model components with parameterizations that may overlap and may even treat twice the same physical process. Ultimately, we could imagine a clearer separation in snowpack schemes where (i) the snowfall density only depends on temperature and humidity (to indirectly represent the variability in falling hydrometeors) and (ii) all the wind-related process are all treated by a well calibrated wind-compaction routine. The data collected in the SLF RWT will improve the wind-compaction routines implemented in the model.
Thanks for this comment! We added this information also in the new paragraph of Section 3.4.2:
L383: "We conclude that a clearer separation in snowpack schemes may improve future model attempts of wind induced snow compaction, where the snowfall density only depends on temperature and humidity (to indirectly represent the variability in falling hydrometeors) and all the wind-related processes are treated by a well calibrated wind-compaction routine."

**Report #2**
**Submitted on 04 Mar 2024**
**Referee #1: Nikolas Aksamit, nikolas.aksamit@uit.no**

The authors have been receptive to my previous concerns and have put in a notable effort to account for them in the updated manuscript. These efforts are appreciated.

Thanks for reviewing again our manuscript and highlighting potential improvements!

The primary concern in my original review was the effect of the wall impacts through the curved section of the ring tunnel. As the authors state "In the curved sections, a large portion of the snow particles are transported along the outer wall due to centrifugal forces. " They have largely addressed these concerns in lines 326-337 (section 3.4.1, not 3.2).

As it stands, I find their response still lacking any transparent quantifications, and a bit confusing in its description:

The authors first describe the snow motion as "sliding" along the wall, which is a very strange process for "large portion of the snow particles" to repeatedly undergo when simulating saltation. Should we now be thinking about coefficients of friction?

Thanks for highlighting this. The word "sliding" is indeed misleading. We updated this section:

L318: "In the curved sections, the particles are transported within a few centimeters distance from the vertical RWT outer wall. The visually identified modes of transport were a mixture of bouncing (saltation), rolling, and sliding along the wall, thus similar to saltation at the horizontal snow surfaces at the straight sections. The particle transport along the curved side walls is inevitable for a compact closed circuit wind tunnel in a cold laboratory of limited dimensions."

The authors then state that the centrifugal forces were 2-3 orders of magnitude smaller compared to forces acting on the particle during impact. How the authors came to this conclusion is not supported by any calculations, or measurements. It is unclear to me where these numbers came from. Did they perform particle tracking around the bend? Just the beginning of the bend or in the middle as well?

Thanks for this comment. We did not perform any particle tracking in the curved sections. We agree that we should be more transparent on where these numbers come from and therefore added the following information:

L322: "The centrifugal forces acting on snow particles in the curved section were estimated being one to two orders of magnitude smaller compared to the forces acting on the snow particles during surface impact while saltating. The maximum centrifugal force was calculated as $F_{c}$ = $m_{p}*v_{p}^2/r = 4.3$ $\mu$N for a large spherical snow particle of 0.5 mm diameter with a mass of $m_{p}$ = 0.06mg, a

maximum horizontal velocity of $v_{p}$ = 6 $ms^{-1}$ ($V_{0.4m}$ $\approx$ 7 $m s^{-1}$) and the RWT radius of the curved section of r = 0.5 m. Horizontal snow particle velocities in snow saltation layers can be approximated as being about 1-2 m$s^{-1}$ lower than the mean horizontal wind speed (Nishimura et al., 2014). The maximum impact force can be calculated as $F_{c}$ = $\Delta E_k$/$h$ = 360 $\mu$N, where $\Delta E_k$ is the kinetic energy difference before and after an impact of a similar snow particle of mass ($m_{p}$ = 0.06mg) estimated from Fig. 5c as $\Delta E_k = 0.5*m_{p}*$ ($V_{in}^2$ – $V_{out}^2$) = 0.5 * $m_{p}$ * 6 $m^2 s^{-2}$ at the same wind speed of $V_{0.4m}$ = 7 $m s^{-1}$. An unknown parameter in this estimate is the height $h$ which defines the particle penetration distance into the snow surface. For small $h$ equal to the particle diameter, the particle impact force is about two orders of magnitude larger than the centrifugal force according to the values above. For increasing penetration distances $h$ (depending on the snow surface elastic or plastic deformation potential), the impact force decreases but is still one order of magnitude larger than the centrifugal force even for $h$ equal 8 times the particle diameter. We conclude that centrifugal forces in the curved section are negligible compared to surface impact forces for our RWT experiments."

Next, the authors suggest the particles are undergoing some sort of bouncing along the wall and estimate the impact angles along the wall are similar to the straight sections, but we don't know where these values come from (numerical simulation? Measurements?). If there are indeed a lot of small hops around the corner, and not some sliding, could you argue the cumulative impact on the crystals is less than during the same transport time in the straight section?

Sorry for the confusion. We assessed the effect of the curvature on the particles based on the above estimation of centrifugal forces. Regarding impact angles in the curved section, we meant the first impact of the particles after the straight test section into the vertical, curved side wall, which is likely the maximum impact angle the particles will experience in the curved section. We did not do any simulation or particle tracking measurements in the curved sections. We modified this Section accordingly:

 L336: "The impact angles of the snow particles first impact into the vertical, curved side walls after a straight section were calculated (based on geometrical considerations) to be within a range of 5°-25°. These angles are comparable to the impact angles $\alpha_{in}$ on the horizontal snow surface in the straight test section (Fig. 4b and d)."

Finally, a stokes number of <0.1 is subsequently provided, as well as the suggestion of "good flow following behavior of the snow particles when the air flow gets redirected in the curved section, resulting in smaller impact angles." Is this to suggest the stokes drag is countering the effect of the curve?

Thank you for this very important question which revealed an erroneous calculation of the Stokes number. Our calculation assumed a low Reynolds number flow which is not the case for our experiments. Therefore, no "good flow following behavior" of the snow particles can be assumed per se reducing impact angles and thus impact forces at the first impact in the curved section. Instead, the flow following behavior strongly depends on the particle size and shape, thus the drag coefficient. Huang et al. (2015) have shown that trajectories of smaller snow particles < 100 µm follow turbulent motions of the flow transitioning into suspension, whereas larger particles > 300 µm have a poor flow following

behavior thus remaining in saltation. Similarly, in our case, impact angles of larger particles are assumed being less reduced than that of smaller particles.

Based on this comment and other comments below, we added new information and revised the entire paragraph:

L335 "Besides the centrifugal forces along the curved side walls, the first impact of snow particles into the vertical, curved side walls after the straight sections introduce additional unnatural mechanical stress on the snow particles, potentially affecting fragmentation. The above introduced estimate of the impact force $F_{i}$ onto the horizontal snow surface is based on impact characteristics determined from the particle tracking measurements, data that is not available for the first impacts at the curved sections. Therefore, we can only provide a discussion of potential differences that may in- or decrease the wall impact force relative to the snow surface impact force. The impact angles of the snow particles' first impact into the side walls were calculated (based on geometrical considerations) to be within a range of 5°-25°. These angles are comparable to the observed impact angles $\alpha_{in}$ on the horizontal snow surface in the straight test section (Fig. 4b and d). The maximum particle impact velocities into the side wall can again be estimated being 1-2 m$s^{-1}$ lower than the mean horizontal wind speed, thus about $v_{p}$ = 5-6 $ms^{-1}$ ($V_{0.4m}$ $\approx$ 7 $m s^{-1}$). These maximum impact velocities are comparable to the maximum impact velocities $V_{in}$ on the horizontal snow surface (Fig. 5a). Geometric vector analysis revealed similar wall normal velocity components for the snow and the curved wall impacts. While the impact angles and velocities are similar, the hard wooden surface of the curved side walls likely increases the impact force relative to the snow surface. Contrarily, the smooth surface of the side walls is assumed to reduce the ejection angle and increase the ejection velocity compared to a snow surface impact, resulting in a decrease of the normalized dissipated impact energy (Fig. 5c) and impact force. The impact angle and the impact force may further be reduced by the particles' ability to follow the flow. Smaller particles (< 100 $\mu m$) have a good flow following behavior (Huang et al., 2015) resulting in a reduction of the impact angles and thus forces. Vice versa, larger particles (> 300 $\mu m$) have a poor flow following behavior resulting in a minor reduction of the impact angle and force. An estimate of the particle size distribution for our experiments (Section 3.6.1, Fig. 12a) reveals that the majority of our snow particles are of a size smaller than 200 $\mu m$, indicating that our particles likely experienced a significant reduction of the impact angle and thus force relative to the impacts analysed based on purely geometrically calculated impact angles. We conclude that these difficult to quantify first particle impacts into the curved side walls after a straight test section introduce some uncertainty but result in similar or in the worst case slightly higher impact forces compared to snow surface impacts. Based on the above discussion, we assume that the mechanical stresses affecting the snow particles in the curved section are comparable to real natural snow transport situations. A more in-depth analysis of the wall-impact forces would require detailed simulations or particle tracking measurements, which is beyond the scope of this work."

Huang, N. and Wang, Z.: A 3-D simulation of drifting snow in the turbulent boundary layer, The Cryosphere Discuss., 9, 301–331, https://doi.org/10.5194/tcd-9-301-2015, 2015.

Does the stokes number change at the curve?

We removed the discussion of the Stokes number due to the reasons discussed in your previous comment.

What are the streamwise/spanwise drag forces and can they account for the acceleration necessary to bend around the curve and reduce wall impact forces?

A reliable estimate of the initial impact forces at the curved side walls after the straight sections would require an in-depth study of the particle size/shape defining drag forces, of the flow field and particle trajectories, of the side wall surface roughness and hardness, ideally using CFD or LES simulations or particle imaging techniques. We argue that this would be way beyond the scope of our work and not necessary at this point as shown in the discussion above.

In addition to clearing up the physical processes in the above narrative you have provided, can you explicitly calculate what the impact forces against the wall are?

Calculating reliable estimates of impact forces would require PIV or LES characterization of the impact/ejection angles and velocities that depend on the surface hardness and roughness.

Why is the amount of force exerted on the particle by the wall during the first impact after the straight section as the particle enters the curve not notable (e.g. when the streamwise velocity, not the vertical velocity, is a potentially major contributor to the particle-wall momentum balance)?

As discussed in the new/revised paragraph, we do not argue anymore that it is not notable.

[Figure]

According to the above illustration, a particle with a horizontal velocity Vh is assumed for both the snow surface and wall impact. The impact velocity Vim,wall is actually lower for the curved wall impact compared to the snow surface impact (Vim,snow). The wall normal velocity Vnormal is in this case 10% lower for the curved wall impact compared to the snow surface impact. However, this analysis considers a perfectly horizontally flying particle for the wall impact. If a vertical component is added to the curved wall impact, the hypothenuse in the red box will be something in between Vh and Vim,snow, while the

angle will only marginally change, and the resulting wall normal impact velocity will become similar to the snow surface impact. Thanks to the small impact angles in the curved sections of max. 25° that are very similar to the snow surface impacts, that fragmentation is not enhanced but similar at these first impacts into the side walls.

We added one sentence to account for this analysis:

L345:" Geometric vector analysis revealed similar wall normal velocity components for the snow and the curved wall impacts."

Presumably an upper bound for this impact can be calculated by using the radius/curvature of the wall to get a maximal first impact angle (that's what we did, resulting in the 5°-25° impact angles) and the maximum particle speed (to get an impact force, we need to know the energy dissipated at the impact which is not measured, and which depends on ejection angle/velocity), and compare that to vertical velocities in the straight section? My concern here is that the horizontal speed is still likely much higher than the vertical and small impact angles may not be sufficient to account for that.

Please see illustration and comments above. The particle impact velocities and impact angles are similar in both cases.

Are the curved wall impact forces 2-3 times smaller than in the straight section because of the curve of the wall and the actual angle of the surface tangent to the curve?

Sorry for the confusion. We found 1-2 (previously 2-3) orders of magnitude smaller centrifugal forces along the entire curved section, not impact forces at the first impact. The range of impact angels (5°-25°) of the first impact was indeed calculated from the surface tangent at the curved section.

Is this where the 25-30 degree (maximum!) impact angle comes from, or is that once the particle is further along in the tunnel?

Yes, please see answers to your comments above. We change to provide the whole range of 5°-25° to highlight that most impact angles are well below 25°.

I don't doubt that the experiment has measured some interesting processes that may be evident in nature, but I still would like the authors to make an effort be quantitatively transparent and rigorous when they argue these curved walls have a negligible effect.

We totally agree and therefore added corrections and more information to be as transparent as possible. Thanks for making us digging deeper into the effects of the curved sections of our RWT, which certainly was necessary and helped to improve the quality and discussion of our results.
Thank you again for allowing me to review this novel and exciting piece of research!

Thank you very much for your valuable time and comments that certainly helped to improve the quality of the article!